# AudSemThinker: Enhancing Audio-Language Models through Reasoning over Semantics of Sound

**Gijs Wijngaard**     **Elia Formisano**     **Michele Esposito**     **Michel Dumontier**
Maastricht University

## Abstract

Audio-language models have shown promising results in various sound understanding tasks, yet they remain limited in their ability to reason over the fine-grained semantics of sound. In this paper, we present AUDSEMTHINKER, a model whose reasoning is structured around a framework of auditory semantics inspired by human cognition. To support this, we introduce AUDSEM, a novel dataset specifically curated for semantic descriptor reasoning in audio-language models. AUDSEM addresses the persistent challenge of data contamination in zero-shot evaluations by providing a carefully filtered collection of audio samples paired with captions generated through a robust multi-stage pipeline. Our experiments demonstrate that AUDSEMTHINKER outperforms state-of-the-art models across multiple training settings, highlighting its strength in semantic audio reasoning. Both AUDSEM-THINKER and the AUDSEM dataset are released publicly[1].

## 1 Introduction

Recent foundation models, such as OpenAI's o1 [54] and DeepSeek R1 [27], have shown that incorporating explicit reasoning phases before generating answers significantly improves performance across tasks. Despite these advances, audio-language models have rarely implemented such structured reasoning approaches. Although general foundation models increasingly separate generation into explicit "thinking" and "answering" stages, audio understanding models still rely largely on end-to-end architectures that directly map audio into text without intermediate reasoning steps. This gap represents a promising opportunity to improve audio understanding capabilities through more deliberate reasoning mechanisms.

Several techniques have emerged to extend test-time reasoning in a controlled and scalable manner. One such method is budget forcing, which inserts "wait" tokens during generation to lengthen the model's reasoning process [52]. Other approaches rely on reinforcement learning to manage a model's thinking budget—the computational effort allocated during inference. For instance, Length-Controlled Policy Optimization (LCPO) [2] jointly optimizes for accuracy and reasoning length, while cosine-shaped reward functions have been used to penalize overly long yet incorrect outputs [11]. Notably, scaling test-time compute has shown significant benefits, with smaller models like LLaMA 3B outperforming much larger ones, such as LLaMA 70B, when allowed more extensive reasoning during inference [4].

A parallel challenge in audio-language modeling is the limited diversity of training data. The field has become highly compartmentalized, with most models trained on overlapping datasets primarily sourced from AudioSet [20] and Freesound [17]. This redundancy is reflected in widely used benchmarks such as AudioCaps [35], WavCaps [50], and various derivatives. For example, studies report that 17.6% of AudioCaps overlap with WavCaps, while Clotho [16] exhibits even higher overlaps - up to 89% - with other datasets [40]. Such cross-contamination contributes to

---

[1] https://github.com/GLJS/AudSemThinker

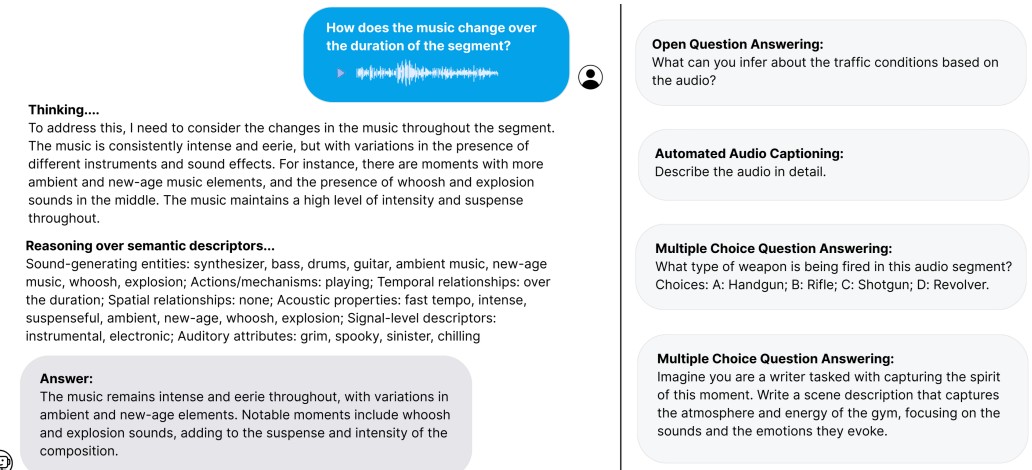

Figure 1: **Schematic overview of the AUDSEMTHINKER model and the AUDSEM dataset**. Left: The AUDSEMTHINKER model's reasoning process, involving a thinking phase, semantic descriptor analysis, and answer generation. Right: Example of four types of tasks in the AUDSEM dataset: open-ended question answering, multiple-choice question answering, audio captioning, and creative writing.

homogenization across datasets, limiting the generalization and robustness of audio-language models [69].

To address the dual challenges of limited reasoning in audio-language models and the lack of diverse training data, we introduce AUDSEMTHINKER (Audio Semantics Thinker), a model that grounds its reasoning process in a structured framework of auditory semantics inspired by human cognition. Specifically, the model analyzes audio in terms of functional components, including sound-generating agents (who), physical sound sources (what), generation mechanisms (how), and contextual cues (when/where). This framework draws on theoretical foundations [18] and experimental research [28, 5] that investigate the perceptual and cognitive mechanisms driving human perception of complex sounds [23]. To support the development of AUDSEMTHINKER, we also construct AUDSEM, a new dataset synthesized from YouTube closed captions, designed to enhance training diversity and reduce overlap with existing resources. Using the AUDSEM dataset, we train AUDSEMTHINKER using two paradigms, Supervised Fine-Tuning (SFT) and Group Relative Policy Optimization (GRPO), and evaluate its performance on a range of established benchmarks, spanning both general audio foundation tasks and audio-based dialogue.

Our main contributions are:

- **AUDSEMTHINKER** — a novel audio-language model that integrates structured reasoning with semantic descriptor analysis, achieving state-of-the-art performance across multiple tasks.

- **AUDSEM** — a newly curated dataset built from YouTube closed captions, specifically designed to reduce overlap with existing datasets. Sound descriptions in AUDSEM are generated via a multi-stage synthetic pipeline, to provide both diversity and high quality.

## 2 Related Work

### 2.1 Audio Datasets with Synthetically Generated Captions

Early audio-language datasets primarily relied on manual annotation or existing metadata. For example, AudioCaps [35] employed Amazon Mechanical Turk workers to generate captions for AudioSet clips. Subsequently, the SAM-S dataset [30] introduced an innovative approach by extracting sound-specific data from closed caption transcripts of Hollywood movies, yielding 116,000 captions. A significant shift occurred with the advent of AI-assisted dataset creation for audio models. In the audio domain, 80% of new datasets created in the first half of 2024 incorporated LLM-generated captions, a marked increase from 69.2% in 2023 [69]. One notable example is Auto-ACD [62], which

employed multiple AI models to process both audio and visual information from VGGSound and AudioSet. The dataset creation pipeline integrated seven different models, including BLIP-2 [42] for image captioning and the CLIP model [57] for categorization. Other examples of AI-assisted dataset creation for audio models include Sound-VECaps [74], which employed CogVLM [66] for visual captioning and LLaMA 3 [26] for generating audio descriptions. This approach produced two complementary dataset variants: Sound-VECaps$_F$, which incorporates detailed visual features, and Sound-VECaps$_A$, which focuses solely on audible content. Building on this, AudioSetCaps [3] introduced a more refined three-stage pipeline that combines large audio-language models for content extraction, LLMs for caption generation, and CLAP-based refinement [70], resulting in 1.9 million high-quality audio-caption pairs.

Our AUDSEM dataset creation methodology advances beyond prior work by integrating audio, video, and YouTube closed caption data through a novel pipeline. This approach employs an ensemble of specialized AI models for multimodal analysis. Table 1 compares our pipeline with other synthetic audio caption datasets.

| Model | Video | Audio | Text | Image | No AS | Think Step | Text Input | Num Models | Length |
|---|---|---|---|---|---|---|---|---|---|
| WavCaps [50] | ✗ | ✓ | ✓ | ✗ | ✗ | ✗ | Metadata | 1 | 44.87 |
| AudioSetCaps [3] | ✗ | ✓ | ✓ | ✗ | ✗ | ✗ | Labels | 4 | 179.20 |
| AF-AudioSet [38] | ✗ | ✓ | ✗ | ✗ | ✗ | ✗ | - | 1 | 143.57 |
| Auto-ACD [62] | ✗ | ✓ | ✓ | ✓ | ✗ | ✗ | Labels | 7 | 103.87 |
| Sound-VECaps$_F$ [74] | ✓ | ✓ | ✓ | ✗ | ✗ | ✗ | Labels | 3 | 180.72 |
| AUDSEM *(ours)* | ✓ | ✓ | ✓ | ✓ | ✓ | ✓ | CC | 9 | 852.63 |

Table 1: **Comparison of synthetic audio caption dataset creation pipelines.** The AUDSEM pipeline uses all four modalities (video, audio, text and image) for its data across nine models. Instead of AudioSet, closed captions were used in its creation. The AUDSEM dataset contains thinking and semantic descriptor steps, and has a higher average character length compared to other datasets.

## 2.2  Audio Models with Reasoning

Several recent efforts have focused on integrating reasoning capabilities into audio-language models. One example is Mellow [12], a lightweight audio-language model that demonstrated strong reasoning abilities. Despite having only 167 million parameters and being trained on 1.24 million examples, Mellow outperformed larger state-of-the-art models across various domains. Another prominent model, Audio-Reasoner [71], was specifically designed for deep reasoning in audio tasks. It introduced a structured reasoning process using a large-scale dataset (CoTA) and a multi-phase "thinking" architecture consisting of planning, captioning, reasoning, and summarization before generating its final response. Audio-CoT [49] explored the integration of Chain-of-Thought (CoT) prompting within large audio-language models, revealing a positive correlation between the length of reasoning paths and model accuracy.

Our work builds on these advances but introduces several key innovations. AUDSEMTHINKER applies a novel semantic analysis framework that decomposes audio into interpretable components—who, what, how, and when/where—enabling more nuanced and human-aligned understanding of audio scenes. Furthermore, it is trained on AudSem, a carefully curated dataset designed to mitigate the data contamination and redundancy issues prevalent in existing audio-language resources.

## 3  AUDSEM Dataset

The AUDSEM dataset was created through a multi-stage pipeline designed to generate high-quality, diverse audio-language data while minimizing overlap with existing resources. The pipeline, visualized in Figure 2, involved several key stages summarized below. A distinctive strength of AUDSEM is its fully automated creation process. While the initial closed captions are manually created by the uploaders of the video on the YouTube platform, the subsequent stages of dataset generation require no additional human intervention. Using YouTube's closed caption subtitles as ground truth for audio events, we implement automatic consistency checks between the generated audio descriptions and the original captions. Misaligned entries are filtered out, providing an implicit quality control

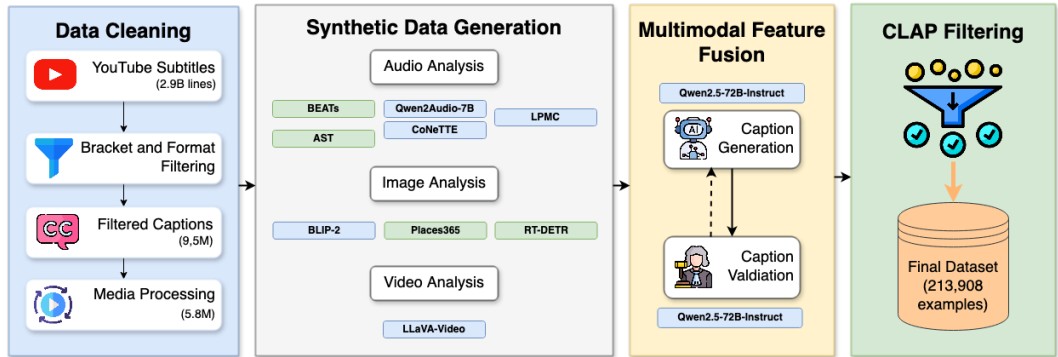

Figure 2: **Pipeline visualization of the creation of AUDSEM.** Models with blue background are generative language models, while models with green background are classification models that predict outputs from a fixed set of possible labels.

mechanism without manual review. This approach enables a highly scalable and resource-efficient pipeline capable of producing high-quality descriptions for millions of audio examples. Our method demonstrates that large-scale, diverse audio-language datasets can be created with minimal human effort while maintaining accuracy and relevance.

The initial dataset creation phase started with a vast collection of YouTube subtitles, filtering for bracketed SDH captions likely representing sound descriptions. The YouTube subtitles were further refined using both BERT [13] classifier and a Mixtral [34] decoder to identify genuine audio events, resulting in 9.5 million potential captions. The corresponding video segments were downloaded, and audio/video streams were standardized (see Appendix B.1 for details).

The following stage involved analyzing the extracted audio-visual segments using multiple specialized models. Audio analysis employed an ensemble including Qwen2Audio-7B [8], BEATs [6], AST [24], CoNeTTE [40], and LP-MusicCaps Model [14]. Visual analysis used BLIP [43], CLIP [57], RT-DETR [48], and Places365 [76] on sampled frames, filtering black frames and deduplicating results. Video analysis utilized LLaVA-Video-7B [75] to capture temporal context from uniformly sampled frames. This stage yielded an initial dataset of 5,332,211 samples, which then underwent multiple filtering steps to ensure quality. Outliers were removed if the cosine distance between their CLAP embedding and the average embedding exceeded 0.9 (for both audio and text). Audio samples shorter than three seconds were also excluded. A final filtering step ensured that the cosine similarity between generated audio captions and original YouTube closed captions was at least 0.5, resulting in 213,908 high-quality samples (see Appendix B.2 for details).

The final stage utilized the Qwen2.5-72B-Instruct model [73] to synthesize the final sound description from the processed multi-modal features. This involved a structured generation approach with enforced JSON output, implementing a thinking phase, an optional semantic descriptor phase, and an answer phase. A separate judging model validated outputs for quality and adherence to guidelines, with regeneration attempts for failed outputs. From the 213k processed examples, four types of tasks were generated (captioning, multiple-choice QA, open-ended QA, creative writing), resulting in approximately 800k-900k final examples depending on the inclusion of semantic descriptors (see Appendix B.3 for details). When comparing against other datasets, there was an overlap of only 12 examples with AudioSet, with one example also being in AudioCaps, and 0 in VGGSound.

Both configurations use XML-style tags to distinguish between phases. In the two-phase approach, the model's output is structured with `<thinking>` and `<answer>` tags. The three-phase approach introduces an additional `<semantic_elements>` section between thinking and answering, allowing the model to explicitly represent key semantic descriptors before generating the final caption. See also Appendix D with Figure 5 for an example of an output from the two-phase approach, and Figure 6 for an example of an output from the three-phase approach.

Figure 3 provides insights into the composition and refinement of the AUDSEM dataset. The left panel details the distribution of broad sound event categories, derived by mapping specific AudioSet labels to their top-level taxonomical parents. Music labels in particular are kept in the filtered dataset. The right panel visualizes the semantic space of captions through PCA-reduced embeddings. It

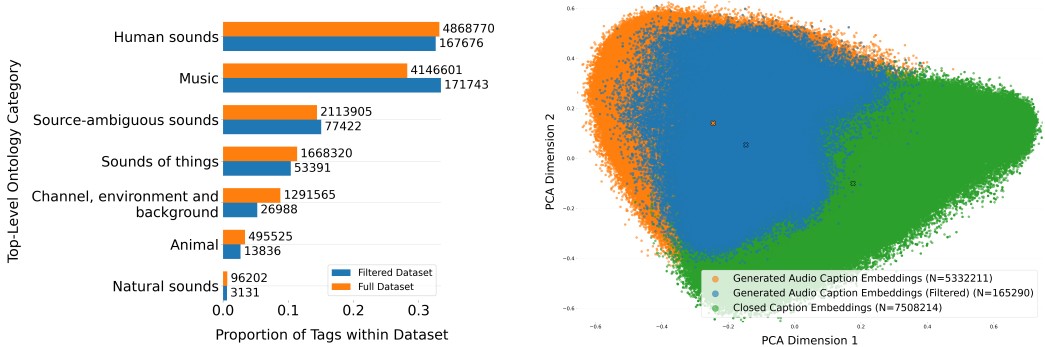

Figure 3: **Visual analysis of AUDSEM dataset characteristics**, showing distributions of sound categories (left) and PCA projection of caption embeddings (right). The filtered dataset was created by filtering out audio caption embeddings with less than 0.5 similarity with the closed captions.

displays the original closed captions sourced from YouTube (green), the audio captions generated by the Qwen2-Audio-7B model (blue), and the resultant filtered samples that constitute AUDSEM (orange). The filtering criterion, a cosine similarity score greater than 0.5 between original and generated caption embeddings, aims to select for high agreement and relevance.

A systematic evaluation was conducted on a randomly selected subset of 100 examples from the dataset, with three annotators assessing both correctness and depth of responses on a Likert scale from 1 to 5. For caption correctness, defined as how accurately the captions describe the events occurring in the audio, the evaluation achieved moderate to substantial inter-annotator agreement across multiple metrics: Krippendorff's Alpha of 0.538, ICC(2,1) of 0.589, and weighted Kappa values ranging from 0.464 to 0.529 using linear weighting. The rating distributions demonstrated strong consistency, with 70-77% of samples receiving the highest rating (5/5) across all three annotators. For response depth, although inter-annotator agreement was lower (Krippendorff's Alpha of 0.207), the ratings remained consistently high, with over 85% of samples receiving the highest score.

## 3.1 Semantic Descriptors

The core idea behind incorporating semantic descriptors into model reasoning is to capture the detailed aspects of audio that a human listener would naturally perceive (see also [23, 68]). Before generating a final caption, the model first engages in a structured "thinking" phase, during which it attempts to identify and describe specific components of the audio signal. This process enables the model to move beyond surface-level recognition toward a more nuanced understanding of what it is "hearing". The descriptors cover both objective and subjective dimensions of auditory perception. Objective descriptors include elements that produce the sound, such as sound-generating agents and physical sound sources, while subjective descriptors address how the sound is perceived, including auditory attributes and non-auditory sensations. This structured reasoning approach allows the model to decompose complex audio scenes into their core semantic descriptors, resulting in more informative and interpretable outputs. See Table 2 for definitions of the semantic descriptors used in our approach.

## 4 AUDSEMTHINKER Model

The proposed approach employs the Qwen2.5-Omni [73] model with 7B parameters. Qwen2.5-Omni is an end-to-end multimodal model that processes diverse inputs including text, images, audio, and video, and generates both text and speech outputs through its Thinker-Talker architecture. For AUDSEMTHINKER, the "Thinker" component, the large language model responsible for text generation, is specifically fine-tuned. This architecture processes multi-modal inputs and generates textual outputs. Fine-tuning occurs exclusively on the AUDSEM dataset (see Section 3); no other fine-tuning is performed. The core idea involves guiding the model to first engage in a "thinking" process, describe the semantic descriptors of the audio, and then produce a final "answer". Two primary training paradigms are explored: SFT and GRPO, both using the AUDSEM dataset, demonstrating the dataset's effectiveness in both scenarios.

| Semantic Descriptor | Description |
|---|---|
| Sound-Generating Agents (Who) | Denotes animated beings that generate sounds, such as people, birds, or animals. Only includes living beings, not pronouns or articles. |
| Physical Sound Sources (What) | Physical objects and substances that generate sounds, typically acting as subjects or objects in sound-related actions (e.g., bells, cars, shoes). |
| Sound Generation Mechanisms (How) | Actions and mechanisms of sound generation, including abstract nouns and verbs describing specific actions or events (e.g., chirping, walking, ringing). |
| Temporal Context (When) | Specific time periods or events providing temporal context for sound generation (e.g., morning, holidays), excluding temporal conjunctions. |
| Spatial Context (Where) | Locations relative to the listener or specific environments where sounds occur (e.g., background, train station, room). |
| Acoustic Surfaces (What/Where) | Physical surfaces and materials that contribute to the acoustic properties of the sound event. |
| Signal Descriptors (Sound Type) | Signal-level acoustic descriptors and basic sound classifications (e.g., noise, chord) without reference to production method, including adverbs that describe sound actions like *loudly*, *softly*, *rhythmically*, and words that characterize the acoustic signal itself such as *buzzing*, *humming*, or *ringing*. |
| Auditory Attributes (Sound Property) | Descriptors of the auditory sensation itself (e.g., loud, soft, steady), excluding source-related adjectives. This includes words that characterize how a sound is heard like *dull*, *hard*, *steadily*, *noisily*, *continually*, *with a ding*, or quantifiers like *repeated 8 times*. |
| Non-auditory Sensation | Non-acoustic attributes and emotional descriptors of perceived sounds (e.g., beautiful, relaxing, calm), including subjective impressions of the sound scene like *quiet* in *quiet bus stop* or *calm* in *calm manner*. |

Table 2: **Semantic descriptors used in the three-phase reasoning approach.**

## 4.1 Supervised Fine-Tuning (SFT)

In the SFT approach, the model is directly trained to produce the desired structured output. We train two primary configurations: AUDSEMTHINKER, trained on the full AUDSEM dataset, and AUDSEMTHINKER-QA, trained exclusively on the subset containing multiple-choice audio question answering examples. For both configurations, the training data is formatted into a conversational structure, including a system prompt, a user prompt containing the audio input and a question (or instruction), and an assistant's response which is the ground truth text containing the `<think>`, optional `<semantic_elements>`, and `<answer>` sections. The loss is computed only on the model completion part, being the assistant's response.

Parameter-efficient fine-tuning is used with LoRA [32], targeting projection layers. Both models are trained using the AdamW optimizer [46] with a learning rate of 2e-04 for one epoch, using a scheduler with linear decay and bf16 precision. Training uses a batch size of four on a single H100 GPU, taking approximately 12 hours for the full dataset (AUDSEMTHINKER) and six hours for the QA subset (AUDSEMTHINKER-QA). A detailed instructional prompt is prepended to the user's question (see Appendix E.3), where the model is guided on the expected output format and reasoning depth (see Appendix 5.3 for a detailed ablation study).

## 4.2 Reinforcement Learning with Group Relative Policy Optimization (GRPO)

Beyond supervised fine-tuning, we investigate the applicability of reinforcement learning to our task and dataset using GRPO [60]. GRPO has shown to be an effective technique in finetuning models and has been used to train state-of-the-art models such as DeepSeek R1 [60] and Qwen2.5 [73]. GRPO allows us to explore whether the AUDSEM dataset is suitable for RL-based finetuning and to specifically test the impact of controlling the model's "thinking budget" during generation. GRPO with Verifiable Rewards (RLVR) [9, 61] is employed, building upon a QWEN2.5-OMNI base model.

We define a set of reward functions designed to guide the model's generation policy towards accuracy, structural adherence, and controlled reasoning length:

- **Accuracy Reward:** This function evaluates the correctness of the content within the `<answer>` tags. For the multiple-choice QA tasks used in GRPO training, it uses string matching to compare the model's selected answer directly against the ground truth choice.

- **Format Adherence Reward:** This reward encourages the model to strictly follow the prescribed XML-tag structure (`<think>`, optional `<semantic_elements>`, `<answer>`). It checks for the presence, correct order, and proper encapsulation of content within these tags.

- **Length Constraint Reward:** Specifically targeting the `<think>` phase, this reward function implements a "thinking budget". It penalizes deviations from a target length for the reasoning content, using parameters to control the penalty strength and tolerance. We adapt the approach from [2] (with $\alpha = 0.1$, $\delta = 0.5$) by adding a reward component for being close to the target length (see Appendix C for details). This encourages the model to generate thinking content near a specified target length while penalizing excessive length.

These reward functions produce scalars for each generation, which are weighted and summed. For the group of generations corresponding to each prompt, the mean and standard deviation of these summed rewards are calculated. An advantage (summed reward $-$ group mean) is computed for each generation, and these advantages are optionally normalized using the group's standard deviation to produce the final scalar reward signal used for policy updates.

For GRPO training, we focus exclusively on the multiple-choice question answering subset of AUDSEM (approx. 140k examples). This decision is driven by the nature of Verifiable Rewards (RLVR), which are most effective when the correctness of an answer can be easily and objectively verified. While typically used for boolean (true/false) verification, RLVR adapts well to multiple-choice questions by treating the single correct option as "true" and the remaining options as "false". This contrasts with open-ended tasks like audio captioning or free-form QA, where verifying the quality or correctness of longer, more subjective generated text is significantly more challenging for an automated reward model.

The resulting model, AUDSEMTHINKER-QA GRPO, is trained with a target thinking budget of 25 words. Similar to SFT, LoRA [32] is used for parameter-efficient tuning. The input prompts for GRPO also include instructions referencing the target thinking length. For training, we use the default GRPO loss type with $\beta = 0.01$, generate six completions per prompt ($k = 6$), and employ a batch size of two per device with bf16 precision. The model is trained on four H100 GPUs for approximately 10 hours, utilizing DeepSpeed ZeRO-3 [58] and vLLM [39] for efficient training and inference.

## 5 Experiments

This section details the evaluation of AUDSEMTHINKER's performance and the AUDSEM dataset's effectiveness. The experimental setup is described first, followed by the main results on established benchmarks and concluding with an ablation study.

### 5.1 Experimental Setup

Evaluation employs the two main training paradigms detailed in Section 4: SFT and GRPO. All models undergo fine-tuning exclusively on the AUDSEM dataset (Section 3). Performance assessment utilizes three established benchmarks:

- **MMAU** (Massive Multi-Task Audio Understanding) [59] (Table 3): Evaluates expert-level knowledge and complex reasoning across speech, environmental sounds and music across 27 diverse tasks. Scores are reported on both Test-Mini (1k clips) and Test (10k clips) sets; some authors only report Test-Mini. MMAU is a multiple choice question benchmark, where the model has to select the answer out of four possible answers. MMAU uses the accuracy metric, as scores are being calculated by the match between the predicted answer and the correct answer. The answer part is extracted out of the model's full response before evaluating it on MMAU.

- **OmniBench** [44] (Table 4): A multimodal benchmark evaluating audio understanding paired with image captions. Models receive both audio and corresponding visual descriptions, answering questions that require integrating information from both modalities. Questions include both multiple-choice and open-ended formats across speech, sound events, and music domains.

| Name | Sound | | Music | | Speech | | Avg | |
|---|---|---|---|---|---|---|---|---|
| | Test-mini | Test | Test-mini | Test | Test-mini | Test | Test-mini | Test |
| Baselines | | | | | | | | |
| Random Guess | 26.72 | 25.73 | 24.55 | 26.53 | 26.72 | 25.50 | 26.00 | 25.92 |
| Most Frequent Choice | 27.02 | 25.73 | 20.35 | 23.73 | 29.12 | 30.33 | 25.50 | 26.50 |
| Human (Test-Mini) | 86.31 | - | 78.22 | - | 82.17 | - | 82.23 | - |
| Pretrained + Supervised Finetuned Models | | | | | | | | |
| GAMA 7B [22] | 41.44 | 45.40 | 32.33 | 30.83 | 18.91 | 19.21 | 30.90 | 31.81 |
| Qwen Audio [7] | 55.25 | 56.73 | 44.00 | 40.90 | 30.03 | 27.95 | 43.10 | 41.86 |
| Qwen2 Audio [8] | 54.95 | 45.90 | 50.98 | 53.26 | 42.04 | 45.90 | 49.20 | 52.50 |
| Mellow [12] | 61.26 | 64.90 | 54.19 | 52.67 | 29.73 | 38.77 | 48.40 | 52.11 |
| Gemini Pro v1.5 [19] | 56.75 | 54.46 | 49.40 | 48.56 | 58.55 | 55.90 | 54.90 | 52.97 |
| Audio Flamingo 2 [21] | 61.56 | 65.10 | 73.95 | **72.90** | 30.93 | 40.26 | 55.48 | 59.42 |
| Gemini 2.0 Flash [55] | 56.46 | 61.73 | 58.68 | 56.53 | 51.65 | 61.53 | 55.60 | 59.93 |
| Kimi-Audio [36] | 61.68 | - | 73.27 | - | **60.66** | - | **65.00** | - |
| AUDSEMTHINKER (ours) | **63.06** | 66.10 | 71.56 | 67.47 | 54.04 | 59.67 | 62.90 | 64.41 |
| AUDSEMTHINKER-QA (ours) | 61.86 | **66.60** | **76.65** | 70.07 | 52.25 | **60.43** | 63.60 | **65.70** |
| Finetuned with Reinforcement Learning | | | | | | | | |
| Audio-Reasoner [71] | 60.06 | - | 64.30 | - | 60.70 | - | 61.71 | - |
| Audio-CoT [49] | 61.86 | - | 56.29 | - | 55.26 | - | 57.80 | - |
| R1-AQA [41] | 68.77 | **69.76** | 64.37 | 61.40 | **63.66** | 62.70 | 65.60 | 64.36 |
| Gemini 2.5 Flash [10] | **73.87** | - | 65.57 | - | 62.16 | - | **67.18** | - |
| Qwen2.5-Omni-7B [72] | 67.87 | - | **69.16** | - | 59.76 | - | 65.60 | - |
| AUDSEMTHINKER-QA GRPO (ours) | 69.67 | 69.20 | **69.16** | **63.13** | 61.26 | **65.77** | 66.70 | **66.03** |

Table 3: **Performance Comparison on MMAU**. Models presented in this paper are marked with (*ours*). Models marked with (QA) are models trained on the subset of the dataset with only audio question answering examples. All our models are only finetuned on the dataset presented in Section 3. All results of others models are directly derived from their respective reports. The best-performing models in each category are highlighted in **bold**, and the second-best scores are underlined.

• **AudioBench** [65] (Table 5): A universal benchmark covering a diverse set of audio tasks, focusing on instruction following capabilities. For our evaluation, we utilize a relevant subset focusing on Audio Question Answering (AQA), including AudioCaps QA [35], Clotho AQA [45] and WavCaps AQA [50]. Automated Audio Captioning (AAC), including AudioCaps [35] and WavCaps [50], and Music Question Answering Music (QA), including MuchoMusic [67]. Qwen2.5-Omni is included in the benchmarks for comparison. The AQA and Music QA benchmarks are open question answering benchmarks, in contrast to MMAU. For AQA, Music QA and AAC we evaluate the full output of the model against a model-as-judge. A Llama 3 model [26] is tasked with evaluating the response against the correct answer.

| Model | Speech | Sound Event | Music | Overall |
|---|---|---|---|---|
| Qwen-Audio-Chat (7B) [7] | 18.39 | 14.66 | 22.64 | 16.64 |
| Reka-core [64] | 26.27 | 28.53 | 29.43 | 23.12 |
| Audio-Flamingo (1.3B) [37] | 24.78 | 26.98 | 21.51 | 23.82 |
| LTU (7B) [25] | 23.12 | 25.42 | 20.00 | 23.91 |
| Gemini-1.5-Pro [19] | 21.02 | 39.82 | 33.96 | 28.02 |
| UnifiedIO2 [47] | 27.15 | 31.13 | 38.87 | 32.49 |
| Qwen2-Audio-7B [†] [8] | 30.48 | **44.82** | 33.02 | 34.23 |
| Qwen2.5-Omni[†] [73] | 33.47 | 37.93 | 40.57 | 35.27 |
| AUDSEMTHINKER (ours) | 32.19 | 34.86 | **51.89** | 34.80 |
| AUDSEMTHINKER-QA (ours) | 34.33 | 35.63 | 40.57 | 35.27 |
| AUDSEMTHINKER-QA GRPO (ours) | **35.29** | 42.00 | 44.34 | **37.51** |

Table 4: **Performance comparison on OmniBench**. Models presented in this paper are marked with (*ours*). The best-performing models are highlighted in **bold**, second best are underlined. † Self-evaluated score, no equivalent score by authors of benchmark exists.

| | AQA | | | Music QA | AAC | |
|---|---|---|---|---|---|---|
| Model | AudioCaps QA | Clotho AQA | WavCaps QA | MuchoMusic | AudioCaps | WavCaps |
| Qwen-Audio [7] | 50.22 | 61.93 | 42.70 | 59.06 | 47.04 | 32.94 |
| MERaLiON [29] | 49.78 | 63.15 | 46.32 | 57.79 | 38.00 | 33.98 |
| Qwen2-Audio [8] | 45.75 | 50.92 | 44.47 | 71.61 | 40.78 | 33.78 |
| Phi4 [1] | 38.47 | 47.87 | 35.13 | 54.42 | 26.39 | 21.88 |
| SeaLLMs [53] | 53.74 | 53.04 | 42.11 | 63.18 | **53.21** | - |
| WavLLM [33] | 29.84 | 43.01 | 26.25 | 44.31 | 5.50 | 6.90 |
| SALMONN [63] | 50.29 | 57.75 | 47.30 | 50.88 | 37.45 | 23.77 |
| Qwen2.5-Omni [72] | 59.62 | 59.91 | 53.29 | 70.09 | 51.97 | **41.48** |
| AUDSEMTHINKER-QA GRPO *(ours)* | 54.19 | 50.85 | 47.70 | 62.00 | 44.06 | 31.92 |
| AUDSEMTHINKER-QA *(ours)* | 58.40 | **68.74** | 31.83 | 74.47 | 39.83 | 31.83 |
| AUDSEMTHINKER *(ours)* | **62.81** | 63.79 | **55.20** | **76.66** | 44.82 | 35.36 |

Table 5: **Performance comparison on AudioBench**. Models presented in this paper are marked with *(ours)*. The benchmarks on AQA, Music QA and AAC in AudioBench were used. The best-performing models are highlighted in **bold**, and the second-best scores are underlined.

## 5.2 Main Results

**GRPO shows promise for specific reasoning enhancements but does not uniformly outperform SFT**. Fine-tuning with GRPO can guide the model towards desired reasoning patterns or thinking lengths. AUDSEMTHINKER-QA GRPO demonstrates a performance increase over its base model, Qwen2.5-Omni-7B, and exhibits less performance divergence across different tasks compared to supervised fine-tuning. However, GRPO may not always surpass SFT, as seen in the music understanding performance of AUDSEMTHINKER-QA GRPO compared to AUDSEMTHINKER-QA. While GRPO can achieve consistent improvements without catastrophic forgetting, SFT appears more effective for substantial gains in understanding novel data, as shown by the larger performance gap between SFT-tuned models (AUDSEMTHINKER-QA, AUDSEMTHINKER) and their base model. RLVR performs well on closed caption benchmarks, but fails to generalize to AudioBench's open audio question benchmark.

**Training dataset composition directly impacts model performance.** AUDSEMTHINKER excels in Music and Sound Understanding tasks, achieving exceptional results in lyrical reasoning (100%), texture interpretation (94.12%), instrumentation (91.43%), and melodic structure (90.91%). This strength stems from the rich music data in AUDSEM and the integration of LP-MusicCaps during dataset creation. AUDSEMTHINKER-QA shows modest improvements on MMAU but not AudioBench, while outperforming competitors on MuchoMusic. The model's weaker speech understanding performance reflects our intended focus on environmental sound and music reasoning, without incorporating speech-to-text models in the data pipeline.

**Length constraint reward impacts performance.** Experiments investigating thinking budget constraints, detailed in Appendix C, demonstrate the effectiveness of the GRPO length constraint reward function (Equation 1, using default parameters $\alpha = 0.1$ and $\delta = 0.5$). The best results on the MMAU Test-Mini benchmark were observed with thinking budgets (25 words, and 100-150 words, see Table 7) that allowed the model to operate within a range where the reward signal was active. Excessively long target reasoning phases, if they fall outside this effective reward range, do not necessarily translate to better performance under the current setup.

## 5.3 Ablation Study

An ablation study was conducted to investigate the impact of three training strategies. We used the MMAU Test-Mini dataset and focused on two primary model architectures: Qwen2-Audio-7B-Instruct and Qwen2.5-Omni-7B. SFT ablations involved training for one epoch on the AUDSEM dataset, taking approximately six hours for QA and 12 hours for the full training, while GRPO ablations were trained for 10 hours (using the checkpoint at step 8400). Ablations were performed over three configurations: Full/QA indicates using all four subsets, the AAC, multiple choice AQA, open AQA, and creative writing data. QA indicates only the multiple choice AQA. Semantic/Simple describes which data configuration is used: with or without semantic descriptors. The pretrained base model is included for comparison.

The key findings from these ablations (summarized in Table 6) indicate that incorporating semantic descriptors generally enhances performance in SFT settings, particularly when fine-tuning on the QA subset. However, this benefit was not consistently observed in the GRPO setting, possibly because the GRPO reward mechanism primarily targets final answer correctness and may not fully leverage intermediate reasoning steps. The use of LoRA for parameter-efficient fine-tuning proved crucial across both SFT and GRPO, significantly mitigating catastrophic forgetting and improving performance. Conversely, training models from scratch results in worse performance.

| Ablation Type | | | Qwen2-Audio | | | | Qwen2.5-Omni | | | |
|---|---|---|---|---|---|---|---|---|---|---|
| Full/QA | Semantic/Simple | LoRA | Sound | Music | Speech | Avg | Sound | Music | Speech | Avg |
| Supervised Fine-tuning | | | | | | | | | | |
| QA | Simple | ✓ | 65.17 | 73.95 | 39.34 | 59.50 | 65.17 | 73.65 | 46.85 | 61.90 |
| QA | Semantic | ✓ | 67.57 | 73.95 | 42.94 | 61.50 | 64.86 | 74.85 | 51.95 | 63.90 |
| QA | Semantic | ✗ | 26.43 | 25.15 | 19.22 | 23.60 | 14.41 | 17.07 | 15.02 | 15.50 |
| Full | Semantic | ✗ | 2.7 | 3.29 | 0.3 | 2.1 | 22.22 | 33.83 | 16.52 | 24.20 |
| Full | Semantic | ✓ | 60.06 | 72.16 | 45.95 | 59.4 | 63.06 | 71.56 | 54.05 | 62.90 |
| Full | Simple | ✓ | 18.32 | 22.46 | 30.63 | 23.8 | 63.66 | 70.36 | 47.45 | 60.50 |
| Group Relative Policy Optimization | | | | | | | | | | |
| QA | Simple | ✓ | 59.76 | 61.98 | 48.65 | 56.80 | 65.47 | 66.17 | 54.35 | 62.00 |
| QA | Simple | ✗ | 54.05 | 56.29 | 42.34 | 50.90 | 52.25 | 52.69 | 44.14 | 49.70 |
| QA | Semantic | ✓ | 60.96 | 61.08 | 49.25 | 57.10 | 63.36 | 62.87 | 56.76 | 61.00 |

Table 6: **Ablation study results on MMAU Test-Mini**. The table compares performance across Qwen2-Audio and Qwen2.5-Omni architectures with 3 ablation configurations.

## 6 Conclusion

In this paper, AUDSEMTHINKER is introduced, a model that advances audio understanding through structured reasoning over semantic descriptors, and AUDSEM, a newly curated dataset designed to minimize overlap with existing resources. We proposed a new pipeline that includes filtering audio captions using the original closed caption to make sure the description still corresponds to the audio. Experimental results show that our model outperforms state-of-the-art approaches across multiple benchmarks, demonstrating particularly strong performance in music-related tasks. Ablation studies indicate that decomposing audio into semantic descriptors boosts performance when effectively integrated with the base models. Additionally, our thinking budget experiments reveal that both short (25 words) and moderately long (100-150 words) reasoning lengths that allow the model to operate within a range within the reward signal yield optimal results. These findings contribute to the field of audio-language modeling by demonstrating that explicit semantic reasoning improves model performance and enables the construction of cleaner, more diverse datasets, helping to address data contamination in zero-shot evaluations.

## 7 Acknowledgement

This work was supported by the Dutch Research Council (NWO 406.20.GO.030 to Prof. Elia Formisano), the Dutch national e-infrastructure with the support of the SURF Cooperative using grant no. EINF-12157, Data Science Research Infrastructure (DSRI; Maastricht University) and the Dutch Province of Limburg. We would like to thank Jopik from Filmot.com for making a dataset of subtitles available for us. We would like to thank Jenia Jitsev and the LAION team for funding this work by providing compute time on both the JUWELS Booster at Jülich Supercomputer Centre (JSC) and Leonardo at CINECA Datacenter.

We gratefully acknowledge the Gauss Centre for Supercomputing e.V. (www.gauss-centre.eu) for funding this part of work by providing computing time through the John von Neumann Institute for Computing (NIC) on the GCS Supercomputer JUWELS Booster at Jülich Supercomputing Centre (JSC), with access to compute provided via LAION cooperation on foundation models and datasets research.

We acknowledge the EuroHPC Joint Undertaking for awarding this project access to the EuroHPC supercomputer LEONARDO, hosted by CINECA (Italy) and the LEONARDO consortium through an EuroHPC Extreme Access grant EHPC-EXT-2023E02-068, provided via LAION cooperation on foundation models and datasets research.

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

# A   Ethics Statement

## A.1   Responsible Data Science

Dataset creation via YouTube scraping introduces a risk of inadvertently including harmful content. Mitigation strategies involved systematically checking and found no terms related to child abuse, hate speech, sexual content, and harassment. YouTube's community guidelines, which prohibit such material, offer an additional safeguard regarding the source data.

Experimental validation involved extensive runs; ablation studies evaluated 62 models 192 times on the MMAU benchmark across various checkpoints and configurations. A performance discrepancy was observed when evaluating models on compute clusters equipped with A100 versus H100 GPUs. The AUDSEM dataset was constructed to minimize overlap with existing benchmarks, facilitating a more candid evaluation. To current knowledge, training data for AUDSEMTHINKER does not include samples from the MMAU benchmark, though certainty is limited due to the obfuscation of MMAU audio filenames. It is acknowledged that the underlying pretrained model Qwen2.5-Omni may have encountered data present in test sets during its initial pretraining.

## A.2   Broader Impact

The scope of this paper is primarily fundamental and does not propose any real-world applications that could benefit society directly. Nevertheless, this research offers potential positive societal impacts by advancing audio-language understanding. Applications could include enhanced audio transcription and captioning for individuals who are deaf or hard of hearing, sophisticated monitoring systems for environmental sounds like avian populations, and automated closed-caption generation for multimedia.

# B   Detailed Data Generation Pipeline

## B.1   Dataset Creation

The dataset creation process begins with a large-scale collection of manually annotated English subtitle data from YouTube videos up until 2024, generously provided by Filmot,[2] a website specializing in searching through YouTube's subtitle repository. This collection approximately 2.9 billion subtitle lines (288GB of text data) from 18,641,150 videos until 31 December 2023.

From these, 75,794,563 lines (2.57% of the total) start and end with brackets, parentheses, or curly braces, excluding dialogue and non-audio related text. This filtering uses regular expressions to identify genuine sound descriptions, following a common convention for Subtitles for Deaf and Hard of Hearing (SDH) captions where sound effects and audio descriptions are enclosed in brackets.

The preprocessing stage incorporates several filtering steps to ensure proper formatting of caption text: (1) Captions longer than 10 seconds and shorter than one second are removed. (2) New lines are removed, quotation marks are standardized and additional filtering removes non-ASCII characters. However, the dataset still contained many subtitles at this point that are not sound descriptions, such as lyrics to songs and narrator descriptions.

Further filtering is performed to ensure the quality of the dataset using two different language models. First, a BERT-based classifier is trained on a manually annotated subset of captions to identify genuine sound descriptions. The training data consists of balanced positive and negative examples, where positive examples represent actual sound effects and negative examples include dialogue, lyrics, and other non-sound descriptions.

A second classification pass uses the Mixtral-8x7B-Instruct model [34] in a zero-shot setting to verify the sound descriptions. The model is prompted to identify whether captions represent genuine audio events based on specific criteria including sound-related verbs, impact descriptions, and audio property descriptions (see prompt in Appendix E.1). Next to a Mixtral-based model, a small sample of 1000 examples of closed caption subtitles from movies and tv shows are utilized to train a simple BERT sentence classification model, which is used to filter out the captions as well. Only captions

---

[2]filmot.com

that are classified as sound descriptions by both models are kept. The remaining dataset contained 9,464,882 captions.

The video segments corresponding to these captions are downloaded using `yt-dlp`, a YouTube video downloader. For each caption, only the relevant audio segment is extracted based on the subtitle timestamps. The download process includes extracting precise time segments using `yt-dlp`'s `-download-sections` parameter before specifying the video ID, to bulk download only the specific segments for each video in a fast manner. The trade-off here is that the resulting extracted video may contain black frames at the beginning of the video. This is a known issue with `yt-dlp` and related tools due to the way YouTube videos are encoded. A part of the dataset contains no video or audio at all, which is removed. Another issue is that some videos are not available for download or are restricted. The remaining dataset contains 6,965,224 video clips, approximately 13.53TB.

Using `ffmpeg`, a tool for video processing, each video is resized to 360p with two frames per second and converted to MP4 format. The audio is converted to WAV format, in 32kHz sample rate with 16-bit depth and one channel. This is done to ensure that the audio and video data are compatible with the models used in the experiments, and also to reduce the size of the dataset and improve the efficiency of the data loading process.

To further improve the data loading process, the dataset is packaged into WebDataset tar files for efficient data loading and processing. In WebDataset, each dataset entry contains paired files, an MP4 audio file pairs with corresponding JSON metadata. Tar files contain 4,096 samples each. The corresponding JSON metadata is uploaded to the HuggingFace Dataset Repository[3].

## B.2    Dataset Processing

The dataset processing pipeline employs three categories of models for comprehensive video analysis: audio models, image models, and a video model. Each model extracts different modalities of information from the video segments.

The audio analysis pipeline utilizes a multi-model ensemble approach to comprehensively analyze the audio content. The audio is first preprocessed by resampling to both 16kHz and 32kHz to accommodate different model requirements. For each audio segment, five specialized models are employed:

1. **Qwen2Audio-7B-Instruct** [8], a large language model fine-tuned for audio understanding, generates detailed natural language descriptions of the audio content. The model processes the 16kHz audio waveform along with a prompt requesting detailed audio description. The model outputs are decoded using beam search with a maximum length of 1024 tokens.

2. **BEATs** (Bidirectional Encoder representation from Audio Transformers) [6], pretrained on AudioSet-2M, performs multi-label audio tagging. The model analyzes the padded audio waveforms and outputs probability scores for audio events. The top five predicted tags and their confidence scores are retained for each segment.

3. **AST** (Audio Spectrogram Transformer) [24], fine-tuned on AudioSet and ESC-50, provides audio event classification. The model processes audio spectrograms and classifies them into predefined categories from the AudioSet ontology. This provides an additional layer of audio event recognition.

4. **CoNeTTE** [40] model analyzes the 32kHz audio to generate contextual audio event descriptions. The model leverages contrastive learning to map audio segments to semantic descriptions of audio events.

5. **LP-MusicCaps Model** [14], a specialized BART-based music captioning model, trained on the LP-MusicCaps dataset, is employed specifically for segments containing music. The model generates natural language descriptions of musical characteristics when music is detected in the audio.

The outputs from all models are combined into a structured format containing the general audio caption, audio tags with confidence scores, event classifications, contextual descriptions, and music-specific captions when applicable. This comprehensive analysis captures different aspects of the audio content, from general descriptions to specific event detection and musical characteristics.

---

[3]https://huggingface.co/datasets/gijs/audsem

For visual analysis, the image processing pipeline employs multiple vision models working in parallel. First, video frames are extracted at regular intervals - for videos under two seconds, frames are sampled at the 1/3 and 2/3 points, while longer videos have frames spread evenly across their duration up to a maximum of four frames. Black frames are filtered out by checking the mean pixel intensity.

The extracted frames are processed through four specialized models:

1. **BLIP** [43], a vision-language model, generates natural language captions for each frame. The model uses beam search with four beams and a maximum length of 20 tokens to generate descriptive captions.

2. **CLIP** [57] performs zero-shot classification against a predefined set of object categories. Each frame is evaluated against text prompts in the format "a photo of a [category]", with the model's logits being converted to probabilities through softmax normalization.

3. **RT-DETR** [48], a real-time object detection model, identifies objects in each frame with a confidence threshold of 0.3. The model's outputs are post-processed to account for varying image sizes while maintaining accurate bounding box coordinates.

4. **A Places365-trained ResNet-50** [76] classifies the scene/environment in each frame. The frames are resized to 256x256, center-cropped to 224x224, and normalized using ImageNet statistics before being processed by the model.

Results from all frames are aggregated using a deduplication strategy that combines identical predictions while preserving their confidence scores. For each unique prediction, scores are averaged across all occurrences using scatter operations on GPU for efficiency. The final output for each video segment contains deduplicated lists of object detections, scene classifications, and generated captions, along with their associated confidence scores.

Video-specific features are extracted using the **LLaVA-Video-7B-Qwen2** model, which processes the temporal aspects of the content. For each video segment, eight frames are uniformly sampled across the duration, with black frames filtered out by checking mean pixel intensity. The frames are preprocessed using the model's image processor. A prompt template is constructed for each video that includes the video duration, number of sampled frames, and their precise timestamps. The model processes these inputs and generates detailed captions up to 2048 tokens in length. The generation uses a deterministic decoding strategy to ensure consistent outputs. The model is compiled using PyTorch 2.0 for optimized performance and processes videos in batches of 32 using multiple worker processes for efficient data loading. The generated captions capture both the temporal progression and spatial relationships between objects and actions in the video segments, providing context that might be missed by frame-by-frame analysis.

The dataset undergoes several filtering steps to ensure quality. From the initial 5,332,211 results, outliers are removed based on embedding distances. The method computes the average embedding using CLAP and measures the cosine distance between this embedding and all items, for both audio and text embeddings. Items with a cosine distance exceeding 0.9 are filtered out. Additionally, the method filters out items where the cosine distance between the generated audio caption and the original YouTube closed caption is below 0.5. This ensures that the audio content aligns with its textual description. Audio samples shorter than three seconds are also omitted to improve model training on substantive audio content. After applying these filtering criteria, the final dataset consists of 213,908 results.

### B.3 Dataset Generation

The dataset generation process combines outputs from the three modality-specific pipelines into a structured format. Initial evaluations of several large language models (Qwen 2.5 [73], QwQ [56], Llama 3.3 [26], Mistral-Small 3 [51] and Deepseek-based models R1 Llama and R1 Qwen [27]) show that Qwen 2.5 performs significantly better, particularly in maintaining consistent JSON output formats. Based on these results, the final dataset generation pipeline uses Qwen2.5-72B-Instruct, implementing structured generation with a schema through the xgrammar library [15] with the help of vLLM [39]. This model with a relatively high number of parameters was chosen due to the tendency that larger models often function as teacher models for smaller student models with knowledge distillation and compression [31]. The generation process follows a two-phase prompt and three-phase reasoning: a thinking phase for reasoning, a semantic descriptor phase for the key

semantic descriptors, and an answer phase for the final caption, both enforced through structured JSON output (see Appendix E.2).

Initial testing identified challenges with generation consistency and thinking trace length. The thinking phase requires detailed reasoning about the audio scene, analyzing primary and background sounds, key events, activities, and the acoustic environment. For this, the prompt requests at least 50 words for the thought phase and a maximum of 50 words for the audio caption. A separate judging model validates outputs by evaluating adherence to generation guidelines. The judge verifies that outputs do not incorporate pre-existing model outputs and maintain focus on audio-relevant information. If an output fails validation, the system attempts regeneration up to five times before skipping the problematic inputs.

With this 212k examples, four different datasets were created for both with and without added semantic descriptors:

1. A audio captioning dataset, where the instruction is "Describe the audio in detail".
2. A multiple-choice audio question answering dataset, where the instruction is generated together with four choices of answers.
3. A open-ended audio question answering dataset, where the instruction is generated together with a text field for the answer.
4. A audio captioning dataset based on creative writing and story-generation.

For audio captioning, a caption was generated for each example. For all other datasets, 2-3 examples were generated for each input. For the full dataset with added semantic descriptors, 797k examples were generated. For the full dataset without semantic descriptors, 873k examples were generated. The split is about 20% for (1), 25% for (2), 50% for (3) and 5% for (4). The total GPU compute time for the dataset generation including preliminary and failed experiments was about 10k hours.

## C  Thinking Budget

To investigate the relationship between thinking length and model performance, we systematically tested different thinking budget configurations. We examined whether allowing the model to perform more extensive reasoning (using more tokens) before generating an answer improves prediction accuracy across different audio understanding tasks. For this experiment, we implemented a controlled approach using GRPO with a maximum length constraint reward function. This reward function, shown in Equation 1, balances correctness with adherence to specified thinking lengths:

$$r(y, y_{\text{gold}}, n_{\text{gold}}) = \begin{cases} \text{clip}(1 - \alpha(n_{\text{gold}} - n_y) + \delta, 0, 1), & \text{if } n_y \leq n_{\text{gold}} \\ \\ \text{clip}(\alpha(n_{\text{gold}} - n_y) + \delta, 0, 1), & \text{otherwise} \end{cases} \tag{1}$$

Where $y$ represents the model's output, $y_{gold}$ is the target output, $n_y$ is the word length of the thinking section, and $n_{gold}$ is the target thinking length. The parameters $\alpha$ and $\delta$ control the penalty strength and tolerance margin, respectively. In our experiments, we set $\alpha = 0.1$ and $\delta = 0.5$.

For the case where the generated thinking length is less than or equal to the target length ($n_y \leq n_{gold}$), the reward function becomes:

$$r(y, y_{gold}, n_{gold}) = \text{clip}(1 - \alpha(n_{gold} - n_y) + \delta, 0, 1) \tag{2}$$

This component becomes zero when:

$$n_{gold} - n_y \geq \frac{1 + \delta}{\alpha} \tag{3}$$

For example, with our parameter values, if $n_{gold} - n_y = \frac{1+\delta}{\alpha} = \frac{1+0.5}{0.1} = 15$, we get:

$$1 - \alpha(n_{gold} - n_y) + \delta = 1 - 0.1 \cdot 15 + 0.5 = 1 - 1.5 + 0.5 = 0 \tag{4}$$

For the case where the generated thinking length exceeds the target ($n_y > n_{gold}$), which corresponds to the original LPCO Max Length Constraint Mode as defined in [2], the reward function becomes:

$$r(y, y_{gold}, n_{gold}) = \text{clip}(\alpha(n_{gold} - n_y) + \delta, 0, 1) \quad (5)$$

This component clips to zero when:

$$n_y - n_{gold} \geq \frac{\delta}{\alpha} \quad (6)$$

For example, with our values, if $n_y - n_{gold} = \frac{\delta}{\alpha} = \frac{0.5}{0.1} = 5$, this term evaluates to:

$$\alpha(n_{gold} - n_y) + \delta = 0.1 \cdot (-5) + 0.5 = -0.5 + 0.5 = 0 \quad (7)$$

This clipping behavior has implications for training when $\alpha$ is 0.1: if the model's average generated thinking length is significantly shorter (e.g., more than 15 words below $n_{\text{gold}}$) than high target lengths like 125 or 150 words, it may consistently receive a zero reward component from this length constraint. This could hinder convergence towards such longer budgets, as the model does not receive a continuous gradient "nudge" from the length reward to increase its thinking length into the desired range. While adjusting $\alpha$ to a much lower value could widen the range before clipping, an overly small $\alpha$ might diminish the length reward's impact, potentially allowing it to be overshadowed by other rewards in the GRPO objective. In experiments, decreasing $\alpha$ was found to be non-effective. It is crucial that the reward function has a slope (before clipping) to ensure effective gradient propagation in GRPO. The referenced Max Length Constraint Mode [2] is designed to apply a soft constraint, gradually penalizing outputs exceeding the target length rather than imposing a hard cutoff, and it incentivizes token efficiency without sacrificing correctness. The $\delta = 0.5$ term helps ensure that correct answers with minor budget violations are still preferred over incorrect ones.

In the experiments, target length parameter $n_{gold}$ was varied in intervals of 25 words, allowing to precisely control the model's thinking budget and measure the impact of reasoning length on performance across various audio understanding tasks.

Table 7 presents the detailed results of our thinking budget experiments across different audio domains. The results show that constraining the model's thinking to specific token lengths has varying effects on performance. Notably, the shortest (25 words) and longer (100-150 words) thinking budgets tend to yield better overall performance compared to mid-range budgets.

| Thinking Budget (words) | Sound | Music | Speech | Total |
|---|---|---|---|---|
| 25 | 69.07 | 70.66 | 59.16 | 66.30 |
| 50 | 64.86 | 67.07 | 58.26 | 63.40 |
| 75 | 65.47 | 66.47 | 55.26 | 62.40 |
| 100 | 67.27 | 67.07 | 60.96 | 65.10 |
| 125 | 67.87 | 67.96 | 58.56 | 64.80 |
| 150 | 66.67 | 67.96 | 60.66 | 65.10 |

Table 7: **Performance across different thinking budget constraints**. Results show accuracy (%) on MMAU Test-Mini dataset for different audio domains when the model is constrained to use specific thinking lengths.

Since the model starts with an average output length in characters of about 95 characters, we see that for models with 125 and 150 words as a constraint, the target is too high and the model is not able to convert to the correct target length. Targets with 25 words and 100 words differentiate just enough from the original length for the length constraint to be effective. As a result, we see that the performance for any model above 100 words is not better than the model with 100 words as a constraint.

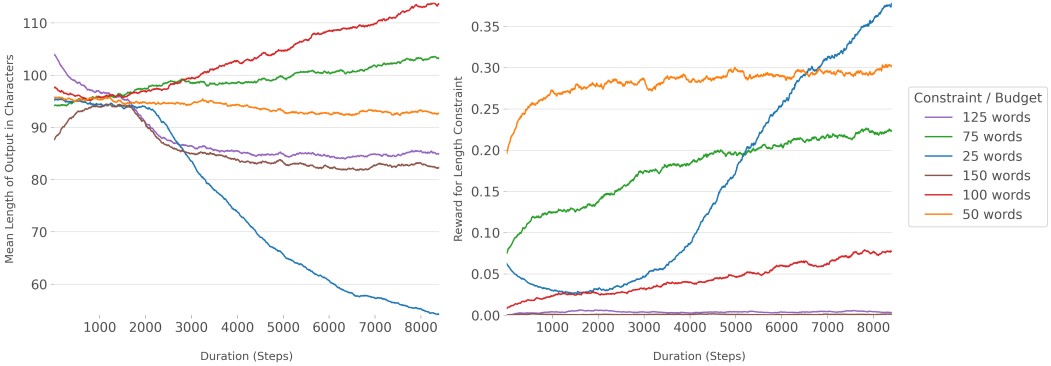

Figure 4: Analysis of thinking budget impact on model performance. Left: How applying a length constraint to the model's thinking nudges the mean length of the output. Right: The relationship between various length constraints and its length constraint reward.

# D    Example Model Outputs

```
<thinking>
Let's think about this. The audio seems to be set in a home environment with a dog and
some background noise. Initially, there's some music playing, but it's not the main focus.
The dog is definitely vocalizing, growling, and barking. There's also a man speaking,
but it's not clear what he's saying. The dog's sounds are the most prominent, indicating
a high level of activity from the dog. The environment feels domestic and warm, which
aligns with the sounds of a pet and a person interacting. The dog's vocalizations suggest
it might be responding to something, possibly the person or another stimulus. The
overall atmosphere is one of interaction and engagement between the person and the
dog, with the dog being quite vocal throughout the segment. The music and speech are
more in the background, adding to the domestic setting but not dominating the scene.
</thinking>
<answer>
A dog is barking and growling in a home environment, with a man speaking in the
background.
</answer>
```

Figure 5: Example of the audio-language model's output (two-phase, without semantic descriptors). The output contains a `<thinking>` section with detailed reasoning about the audio content, followed by an `<answer>` section with the concise caption.

```
<thinking>
I start by considering the primary sound, which is a seal roaring. This is a distinct,
powerful sound that stands out in the audio. The roar is followed by a series of grunts,
which add to the intensity of the scene. In the background, I hear intermittent wind noise,
which suggests an open, coastal environment. There are also some bird sounds, which
are not as prominent but add to the natural setting. The overall sound is consistent with
what you might expect in a beach environment with a large seal population. I also note
some background sounds that could be vehicles or trains, but these are less noticeable
and do not seem to be the main focus of the audio. Overall, the primary sound is the
seal roaring, which is the most significant and noticeable element in the audio.
</thinking>
<semantic_elements>
1. Sound-generating animated beings with descriptive adjectives:
- Seals roaring and grunting (powerful, intense)
2. Physical objects/substances generating sound:
- Birds (singing)
3. Actions/mechanisms of sound generation:
- Roaring and grunting (seals)
4. Temporal context if present:
- Continuous roaring followed by grunts
5. Spatial context and environment:
- Open, coastal environment with seals on a beach
6. Acoustic surfaces and materials contributing to the sound:
- Sandy beach, open air
7. Signal-level sound descriptors:
- Powerful, deep roaring
- Intermittent, soft grunting
8. Auditory sensation attributes:
- Deep, resonant roaring
- Intermittent, low-pitched grunting
9. Subjective/emotional descriptors:
- Intense, powerful, natural
</semantic_elements>
<answer>
A seal roars powerfully, followed by deep grunts, in a coastal environment with wind
and birds in the background.
</answer>
```

Figure 6: Example of the audio-language model's output (three-phase). The output contains a `<thinking>` section with detailed reasoning about the audio content, followed by an `<semantic_elements>` section with the key semantic components, and then an `<answer>` section with the final caption.

# E   Prompts

## E.1   Filtering Prompt

> **Prompt for Caption Filtering**
>
> You are a friendly chatbot whose task it is to filter out bad data. You will get a closed caption corresponding to a video clip. Your task is to state whether the caption is a correct subtitle for deaf or hard-of-hearing people. Correct captions in this task are those that correspond to words that could represent an actual sound being made. This could either include a verb that states an impact or sound types or properties like "sound", "noise" or "music". Incorrect closed captions include sentences that someone is saying in the video clip, or sentences that are not related to the video clip at all. All captions are in English. All captions are within curly brackets or square brackets []. Examples of correct captions include: - "(laughs)" or "(laughter)" - "[XBOX SOUND]" - "[chicken bocking imitation]" - "(cereal grains smacking onto wood)" - "(collision)"
> Examples of incorrect captions include: - "[ transport ]" - "(Wishes are left to wither by time.)" - "(look, I like my nightmareless sleep; I'll play some scary games when I feel too peaceful)" - "[A calm navy color] [TinyTAN character detail]" - "[Haotian Sword Tower]"
> Is the following caption correct? Please only answer "yes" or "no"

## E.2   Caption Generation Prompt

> **Prompt for Judging Captions**
>
> You are a strict judge for an audio caption generator. Your task is to verify whether the generated output adheres to all the rules from the original prompt. In particular, check the following: 1. The 'thinking' process should contain a Chain-of-Thought (CoT) reasoning process. 2. The 'thinking' process must not mention "predictions per second" or any similar phrasing. 3. The 'thinking' process must not include any of the original data fields directly. 4. The 'answer' should be a valid audio caption containing no visual elements or contexts.
> Examine the generated output below carefully and respond with a JSON object that includes: - "valid": set to true if all rules are followed, or false if any rule is broken. - "reason": if false, a brief explanation of which rule(s) were violated.
> — Generated Output Start — generated_output — Generated Output End —

## Prompt for Caption Generation

You are an expert audio caption generator to create training data for an audio model. Your task is to create a detailed caption that describes what happens in an audio segment, including a Chain-of-Thought (CoT) reasoning process. You will be provided with various types of information extracted from audio processing models and supporting visual context. Your goal is to write a thinking process and answer as if you would only have the audio itself, without any of the following information.

Given Information: 1. Basic Information: - Video ID: video_id - Time Segment: start to end seconds - Original Closed Caption: text (This is the most important information to keep in mind)

2. Model-Generated Audio Information: - Audio Caption: audio_caption - Audio Tags (each with a confidence score): audio_tags - Short Audio Caption: conette_candidates - Predictions Per Second (key is the second, value is dict of sound and confidence score): sat_predictions music_caption_section

3. Supporting Visual Context: Scene Description: caption Detected Objects (COCO labels): objects Scene Classification (Places365): places

Context Evaluation Guidelines: 1. Use visual information ONLY if it: a) Strongly aligns with AND confirms audio evidence b) Provides essential acoustic environment context unavailable from audio 2. Ignore visual information if: a) Contradicts audio evidence b) Talks about text/graphics/static images c) Describes visual-only elements NEVER mention the visual context or visual elements in the thinking step, ONLY use it to infer the audio context.

In your output in the thinking step, analyze the audio scene in detail, reason about the primary and background sounds, and describe what happens in the audio. Include key events and activities, and the environment and context.

Guidelines: - Use natural, descriptive language - The thinking should be at least 50 words - Keep the final caption under 50 words - Do not include timestamps - Do not mention specific speech content unless crucial to understanding the audio scene - Use the prediction per second to determine the order of the sounds in the caption - The reasoning process MUST include thought expressions in natural language. This includes discourse markers, hesitation phrases, cognitive markers and casual suggestions. - NEVER describe or mention the original data fields directly in your reasoning process. You are generating training data for an audio model, and the model should learn to reason from the audio itself and NOT from the extracted data given including any of the visual context.

DO NOT mention any of the outputs of the models in the thinking step.

Please provide your response as a JSON object with the following keys: - thinking: The thinking process - answer: The caption

### E.3 Prepended Prompt on Model Generation

## Prompt for Generation with Semantic Descriptors

You are given a question and an audio clip. Your task is to answer the question based on the audio clip. First, think about the question and the audio clip and put your thoughts in <think> and </think> tags. Then reason about the semantic elements involved in the audio clip and put your reasoning in <semantic_elements> and </semantic_elements> tags. Then answer the question based on the audio clip, put your answer in <answer> and </answer> tags.

## Prompt for Generation without Semantic Descriptors

You are given a question and an audio clip. Your task is to answer the question based on the audio clip. First, think about the question and the audio clip and put your thoughts in <think> and </think> tags. Then answer the question based on the audio clip, put your answer in <answer> and </answer> tags.

