# OpenReview forum: "AudSemThinker: Enhancing Audio-Language Models Through Reasoning over Semantics of Sound"
_NeurIPS.cc/2025/Conference — NeurIPS 2025 poster_

### Official Review · Reviewer_1TFH · 2025-06-18

**Clarity:** 3
**Significance:** 3
**Originality:** 2
**Rating:** 4
**Confidence:** 4

**Summary:**

This paper proposes AudSemThinker, a new audio-language model designed for structured reasoning over fine-grained auditory semantics. To support this, the authors introduce AudSem, a dataset curated for audio semantic reasoning tasks. The dataset is built using a multi-stage pipeline that filters caption-audio pairs to minimize data contamination, aiming to preserve zero-shot integrity. The proposed model shows improved performance across several tasks and is especially strong in music-related scenarios. Ablation studies demonstrate the utility of semantic decomposition and control over reasoning length (“thinking budget”).

**Questions:**

- Please clarify the role of audio in the tasks. Are there examples where the model must truly listen to the audio to answer the question, or can it succeed via language priors alone? Including such examples would help assess audio dependency and reasoning depth.

- Show dataset examples and quality checks. Manual inspection results (e.g., sample correctness, audio-caption match) would greatly increase confidence in the dataset's usefulness.

- Update baselines. Why is Gemini 2.0 Pro omitted? It is publicly available and more relevant than Gemini 1.5 Pro. Comparing with stronger baselines like Qwen2.5-Omni is essential if you are claiming SOTA.

- Include harder reasoning benchmarks. Can you evaluate your model on MMAR or OmniBench? These tasks involve multistep reasoning and would more rigorously validate your model’s claimed strengths.

- Revise related work. Include recent, competitive Omni-style models and discuss how your approach differs or complements them in terms of reasoning ability.

**Ethical Concerns:**

["Major Concern: Data privacy, copyright, and consent"]

**Final Justification:**

All concern has been address and I appreciate authors' efforts

**Limitations:**

No detailed discussion on limitations. Besides, the proposed dataset should include a descrioption on copyright and ethic issue on the YouTube audio.

**Quality:**

3

**Strengths And Weaknesses:**

Strengths
- Timely problem focus: The paper addresses a meaningful challenge in audio-language modeling: moving beyond perception into structured semantic reasoning.

- Dataset construction: The paper attempts to mitigate data contamination, a known issue in zero-shot evaluation, which is commendable.

- Model design and ablation: The use of semantic descriptor decomposition and thinking budget analysis provides some insight into model behavior and optimization.

- Performance improvement: The proposed method outperforms selected baselines under the presented settings.

Weaknesses
- Incomplete dataset documentation:

- - Section 3 only includes 3.1, making the subsection label unnecessary.

- - There is no manual validation or dataset example shown, leaving the actual quality of the generated dataset unclear. It's unclear whether the questions require reasoning, perceptual matching, or are language-biased. Are audio inputs essential or optional?

- Omission of key related work: The Omni-style models like Qwen-Omni and Qwen2.5-Omni are not cited or compared, even though they currently surpass older models like Audio-CoT in reasoning capabilities.

- Outdated baseline: Gemini 1.5 Pro, used as a baseline, was deprecated well before NeurIPS submission deadlines. Newer versions like Gemini 2.0 Pro have been shown to outperform many models, including the proposed one.
 The claim of "state-of-the-art" is misleading without up-to-date comparisons.

Benchmark scope is limited:
- The proposed model is not tested on more challenging reasoning benchmarks, such as MMAR (multi-step audio reasoning) or the OmniBench reasoning questions with image captions.
Without these evaluations on more difficult reasoning tasks, it's hard to gauge true reasoning capabilities under generative or compositional settings.

---

> ### Author Rebuttal · Authors · 2025-07-31
>
> We would like to thank the reviewer for their thoughtful and constructive feedback. We appreciate the time and effort spent in evaluating our work. We agree with the assessment of the strengths and weaknesses and have addressed the concerns raised as follows:
>
> > Section 3 only includes 3.1, making the subsection label unnecessary. There is no manual validation or dataset example shown, leaving the actual quality of the generated dataset unclear. It's unclear whether the questions require reasoning, perceptual matching, or are language-biased. Are audio inputs essential or optional?
>
> We thank the reviewer for pointing out the subsection labeling issue, which we will correct in the final version. Regarding dataset examples, we have included them in Figures 5 and 6 in Appendix section E. We went over around 100 examples ourselves to validate the data, but did not do a systematic review. Our pipeline is designed to minimize human intervention, making it efficient to use for various data collection applications. This is a key selling point of our approach, which is why we opted for an optimized pipeline.
> All items in our dataset include \<thinking\> and \</thinking\> tags before the \<answer\> tags, requiring models to reason before answering questions. Each question involves only one audio input, and both audio and language are essential components. Models trained on our dataset must process both the question text and audio input to generate correct responses.
>
> Our dataset consists of about 20% audio captioning samples, 25% of multiple choice questions, 50% of open-ended questions and 5% examples in a creative storytelling task. For details, see Appendix C.3.  Example questions include:
> "What's happening in this audio? Describe it thoroughly."
> "What additional sound can be heard in the background alongside the male voice and music?"
> "What instruments can be heard in the music, and what do they add to the composition?"
> For more examples, the AudSem datasets are released publicly on HuggingFace Datasets platform for easy access and download. Due to NeurIPS anonymity requirements, we cannot provide the direct link in this response.
>
> To validate the necessity of audio inputs, we evaluated the multiple choice questions using Qwen 3 (a model capable of processing only text inputs) against our AudSemThinker model. On a held-out subset of 247 examples of the multiple-choice audio question answering task not used in training, Qwen 3 scored 47.37% while AudSemThinker achieved 70.45%. Random guessing would yield 25% accuracy on this four-choice task.
>
>
> > Omission of key related work: The Omni-style models like Qwen-Omni and Qwen2.5-Omni are not cited or compared, even though they currently surpass older models like Audio-CoT in reasoning capabilities.
>
> We appreciate this observation. We did compare Qwen2.5-Omni in our experiments (Table 3), where it achieved 67.87% on sound, 69.16% on music, and 59.76% on speech categories. As it is a general foundation model, we categorized it separately from the "Audio Models with Reasoning" section, which focuses on specialized models like Audio-Reasoner and Audio-CoT. We also included Qwen2.5-Omni in Table 4 for AudioBench performance. We will clarify this distinction in the revised manuscript.
>
>
> > Outdated baseline: Gemini 1.5 Pro, used as a baseline, was deprecated well before NeurIPS submission deadlines. Newer versions like Gemini 2.0 Pro have been shown to outperform many models, including the proposed one. The claim of "state-of-the-art" is misleading without up-to-date comparisons.
>
> We acknowledge this valid concern. The Gemini 1.5 Pro benchmark results were taken directly from the MMAU leaderboard, as Gemini 2.0 Pro had not yet been evaluated on this dataset by the benchmark owners. Our AudSemThinker model outperforms the best Gemini Pro 2.0 model available in all categories:
>
>
> | Model | Sound (Test-mini/Test) | Music (Test-mini/Test) | Speech (Test-mini/Test) | Total (Test-mini/Test) |
> | ----- | ----- | ----- | ----- | ----- |
> | Gemini 2.0 Pro Flash | 56.46/61.73 | 58.68/56.53 | 51.65/61.53 | 55.60/59.93 |
> | AudSemThinker | 63.06/66.10 | 71.56/67.47 | 54.04/59.67 | 62.90/64.41 |
> | AudSemThinker-QA | 61.86/66.60 | 76.65/70.07 | 52.25/60.43 | 63.60/65.70 |
> | AudSemThinker-QA GRPO | 69.67/69.20 | 69.16/63.13 | 61.26/65.77 | 66.70/66.03 |
>
>
>
> We will update our comparison claims accordingly in the revised manuscript.
>
> > The proposed model is not tested on more challenging reasoning benchmarks, such as MMAR (multi-step audio reasoning) or the OmniBench reasoning questions with image captions. Without these evaluations on more difficult reasoning tasks, it's hard to gauge true reasoning capabilities under generative or compositional settings.
>
> We agree that additional evaluations on reasoning benchmarks would strengthen our paper. Currently, our model has been evaluated on the test-mini and test benchmarks of MMAU (Table 3), as well as AudioCaps QA, Clotho AQA, WavCaps QA, MuchoMusic, AudioCaps, and WavCaps (Table 4).
> Regarding the specific benchmarks mentioned, the MMAR benchmark was released on May 19, after the NeurIPS deadline of May 15. Regarding OmniBench, our model specializes in audio-language understanding and was not designed to process the visual inputs required by this benchmark, making it outside our current evaluation scope. We will consider evaluating additional audio reasoning benchmarks in future work and clarify these limitations in the revised manuscript.
>
> > No detailed discussion on limitations; the dataset should include a description of copyright and ethical issues regarding YouTube audio.
>
> We provide dataset limitations in both the main paper and the appendix, including weaker speech understanding, the performance benefit of semantic descriptors in GRPO training, performance discrepancy across GPUs, thinking budget constraints, and possible data contamination in the underlying pretrained model.
> Regarding copyright and ethical issues, the YouTube data was gathered using open material freely available on the internet, and the dataset is shared under the CC-BY-NC-SA-4.0 license. We will expand this discussion in the revised manuscript to address the reviewer's concerns more comprehensively.
>
> We appreciate and would like to thank the reviewer for their constructive feedback.

---

> ### Comment · Reviewer_1TFH · 2025-08-06
>
> Thank you for the experiments and explanation. And I would like to share two more concerns:
> - Can you provide the statistics and case studies on the 100 samples you went over? Given the dataset is annotated by a total of 9 different classification and generative language models across 3 modalities (Image, Audio, and Video) to provide the LLM with diverse perspectives for generating captions, there might be some system biases such as the language bias in the LLM backbone pre-training corpuus, or the bi-modality understanding corpus, or the unsatisfying capability of long context . A detailed evaluation on such 100 sample is necessary to elabourate the quality of dataset.
> - The held-out evaluation subset of 247 multiple-choice audio question-answering examples reveals a severe language bias in the proposed dataset. Notably, Qwen3—a model that does not access audio—achieves 47.37% accuracy, far above the random baseline of 25%, indicating that many questions can be answered through language priors alone. This undermines the core premise that the task requires auditory understanding and raises concerns about the dataset's validity for training and evaluating audio-language reasoning models. The authors are strongly encouraged to analyze the source of this bias—whether stemming from the LLM backbone, the caption generation process, or insufficient audio grounding—and implement corrective measures, such as [1] and [2], to ensure that audio input is genuinely necessary for task success. Without such steps, the dataset risks overestimating model capabilities and failing to advance the field of true multimodal reasoning.
> - Gemini 2.5 Pro recently demonstrate a much better results on audio reasoning and speech related tasks compared to Gemini 2.0 Pro and it would be helpful to see its performance. Besides, it is not clear which model does the author refer to by "Gemini 2.0 Pro Flash". Gemini 2.0 Flash and Gemini 2.0 Pro are both powerful multimodal models, but they serve different purposes. Gemini 2.0 Flash is designed for speed and real-time applications, while 2.0 Pro offers more advanced reasoning and is better suited for complex tasks and enterprise solutions. So 2.0 Pro and 2.5 Pro are reasoning close-source SOTA of this task
> - While scope of Omnibench is omni-modality reasoning, the paper has different setting with image caption and audio transcripts, making it possible to evaluate bi-modality LLMs other than Omni modality LLMs.
>
> [1] TemporalBench: BENCHMARKING FINE-GRAINED TEMPORAL UNDERSTANDING FOR MULTIMODAL VIDEO MODELS
>
> [2] Are You Really Listening? Boosting Perceptual Awareness in Music QA Benchmarks

---

> > ### Author Response · Authors · 2025-08-07
> > **Response to Official Comment by Reviewer 1TFH**
> >
> > We thank the reviewer for their valuable comments and like to address each question below:
> >
> > > Can you provide the statistics [...]
> >
> > We conducted a systematic manual evaluation of 100 randomly selected examples from our dataset, which encompasses four distinct subsets. The evaluation protocol assessed two key dimensions: factuality and depth. Factuality measured the factual correctness of generated questions, thinking processes, and answers with respect to the audio content using a Likert scale of 1-5. Depth assessed the thoroughness and level of detail in the captions relative to the audio content, also using a Likert scale of 1-5.
> >
> > Our results showed that 74% of samples achieved perfect accuracy (5/5), 11% were highly accurate (4/5), 9% were moderately accurate (3/5), and 3% contained significant errors. For 3% of samples, assessment was inconclusive without additional context. In terms of depth, 97% of samples demonstrated sufficient level of detail. This evaluation was conducted using the Argilla annotation platform.
> >
> > Regarding concerns about system biases, we specifically designed our annotation pipeline to leverage a statistical averaging effect by utilizing 9 different models to allow our caption-generating LLM to consider multiple perspectives and incorporate the most coherent elements from each model's output. We purposefully selected the best performing 72B open-source model at that time to maximize the quality of the synthesized captions.
> >
> > We implemented an additional LLM review process to address biases with a “judge” that assessed each caption on multiple quality dimensions, including ensuring the absence of visual elements or contexts not present in the audio.  Additionally, we filtered our dataset by comparing generated captions against a “gold standard”: the original human-annotated closed captions from the YouTube videos, and ensured alignment between generated captions and original annotations with an embedding model.
> >
> > > The held-out evaluation subset [...]
> >
> > We appreciate the reviewer highlighting this important concern regarding language bias in our dataset. This phenomenon appears to stem from the dual nature of our dataset questions, which intentionally incorporate both language reasoning and audio comprehension components. Questions such as “How does the thunder sound in the audio segment?” or “What might the cough at the end indicate about the environment?” provide contextual cues that language models can leverage, even while full answering still requires audio processing. In multiple-choice settings, these contextual cues may enable language-only models to make educated guesses that exceed random chance.
> >
> > The work by Rouditchenko et al. [1] provides valuable context for this observation. Their research demonstrates that fine-tuning a model without audio on a text-only dataset improved audio-based performance, achieving 49.3% on the new MMAU benchmark during inference without audio, and 51.7% when trained on a dataset without audio. While models fine-tuned with audio still substantially outperform those without, their findings align with our observation that models like Qwen3, trained with RL techniques, can develop surprising proficiency at answering audio questions without audio input.
> >
> > [1] Omni-R1: Do You Really Need Audio to Fine-Tune Your Audio LLM?
> >
> > > Gemini 2.5 Pro recently demonstrate [...]
> >
> > We acknowledge this mistake. We evaluated on “Gemini 2.0 Flash”. We obtained the naming “Gemini Pro 2.0 Flash” from the MMAU benchmark website and copied that result. We were unable to evaluate on Gemini 2.0 Pro, as it does not exist in the Google API anymore and was only released to a select number of customers at the time of release.
> >
> > We acknowledge Gemini 2.5 Pro's strong performance on audio reasoning and speech-related tasks. Our comparative analysis shows that Gemini 2.5 Pro achieved 67.2% on the MMAU benchmark, while our best model reached 66.7%. In category-specific performance, Gemini 2.5 scored 73.87% on sound, 62.16% on speech, and 65.57% on music. Our model scored 69.67% on sound, 61.26% on speech, and 69.16% on music. Notably, our model outperforms Gemini 2.5 Pro in the Music category. Additionally, our open-source model outperforms Gemini 2.5 Flash (64.3% overall), which we believe is comparable in size to our model.
> >
> > > While scope of Omnibench [...]
> >
> > We evaluated our AudSemThinker-GRPO model on the OmniBench benchmark in the Image(Text) & Audio category, achieving 39.77% overall, with 37.36% on sound, 44.34% on music, and 35.29% on speech. This positions us fourth on OmniBench overall and second to models of similar size, outperforming models such as Gemini-1.5-Pro and Baichuan-Omni-1.5.
> >
> > We identified challenges in OmniBench with examples containing visual textual clues (e.g., “What are men doing?” with answer options referring to clothing). Such multimodal inference is inherently difficult for our model, which was specifically designed to process audio data only.

---

> ### Comment · Reviewer_1TFH · 2025-08-08
>
> Thank you for the reply.
>
> While the authors report a manual check of 100 samples, it remains unclear whether annotator outputs show statistical consistency across different annotators are similar. Such analysis is essential to quantify potential systematic biases in the pipeline.
>
> Moreover, the 47.37% no-audio accuracy suggests that, in many cases, two answer choices can be eliminated purely via language priors—potentially one due to perceptual errors, one due to reasoning errors, and the last one due to both errors —yielding similar accuracy even without true audio grounding. This raises the possibility that models with weak reasoning but strong perceptual matching (or vice versa) could attain comparable scores, undermining claims of audio-dependent reasoning. But this assumption need further case study confirm and address this bias if any.
>
> If the consistency among all annootators are high and the case study indicate a high quality of the proposed dataset, I could change my decision on this paper.

---

> > ### Author Response · Authors · 2025-08-09
> >
> > We thank the reviewer for their thoughtful response and for investing time and effort in evaluating our dataset with the Qwen3 model, as well as for engaging constructively in this discussion.
> >
> > > While the authors report a manual check of 100 samples, it remains unclear whether annotator outputs show statistical consistency across different annotators are similar. Such analysis is essential to quantify potential systematic biases in the pipeline.
> >
> > We conducted a systematic evaluation with three annotators on 100 randomly selected examples, to assess both correctness and depth of responses on a Likert scale from 1 to 5.
> >
> > For correctness of the captions in terms of how well they describe the events happening in the audio, we achieved moderate to substantial agreement across multiple metrics: Krippendorff's Alpha of 0.538, ICC(2,1) of 0.589, and weighted Kappa values ranging from 0.464 to 0.529 using linear weighting. The rating distributions show strong consistency, with between 70-77% of samples receiving the highest rating (5/5) across all three annotators. For depth of responses, while agreement was lower (Krippendorff's Alpha of 0.207), the ratings remained consistently high with almost all samples (85%+) receiving the highest score.
> >
> > We agree with the reviewer that a larger manual analysis would have been beneficial to check for potential algorithmic biases, but we would like to remark that our goal was to develop a fully automated pipeline, which was crucial for demonstrating the scalability of our approach.
> >
> > > Moreover, the 47.37% no-audio accuracy suggests that, in many cases, two answer choices can be eliminated purely via language priors—potentially one due to perceptual errors, one due to reasoning errors, and the last one due to both errors —yielding similar accuracy even without true audio grounding. This raises the possibility that models with weak reasoning but strong perceptual matching (or vice versa) could attain comparable scores, undermining claims of audio-dependent reasoning. But this assumption need further case study confirm and address this bias if any.
> >
> > We view this as reflecting an important aspect of audio-language understanding rather than a fundamental flaw, as this is a widespread phenomenon in the field. Audio-language models fundamentally differ from pure audio classification models in that they must integrate both audio perception and language reasoning. The language component enables models to understand context, interpret prompts, and formulate coherent responses, which are capabilities that have become remarkably sophisticated in modern language models. We believe this explains why performance exceeds the 25% random baseline even without audio input.
> >
> > However, we emphasize that our dataset extends far beyond multiple-choice questions, which constitute only 26.72% of AudSem (213k examples) and 24.10% (211k) of AudSem-Simple. The remaining dataset together includes 376k and 473k open-ended question-answering pairs, 163k and 161k audio captioning examples, and 46k and 29k creative writing examples, all requiring natural text generation where language biases from multiple-choice options are eliminated. These tasks require captioning metrics for evaluation and present substantially harder challenges where audio grounding becomes essential. This diverse composition ensures that models trained on our dataset must develop genuine audio understanding capabilities rather than relying solely on language priors.

---

### Official Review · Reviewer_nCca · 2025-07-03

**Clarity:** 3
**Significance:** 4
**Originality:** 3
**Rating:** 4
**Confidence:** 4

**Summary:**

The paper introduces AudSemThinker, an audio language model that performs structured reasoning over semantic audio descriptors inspired by human cognition. It also introduces a novel dataset called AUDSEM curated from YouTube closed captions to reduce overlapping with existing audio datasets which are mainly sourced from AudioSet and Freesound. The proposed model is trained on AudSem on both SFT and GRPO training paradigms. AudSemThinker acheives strong results on MMAU and AudioBench datasets.

**Questions:**

1. Could you elaborate more on how captions of larger audios are generated?
2. I am curious how is the performance on music dataset so good as the distribution of music in the dataset characteristics chart shows the dataset contains more of sound events. I would also be interested to know more about the dataset distribution for sound events. Did you filter out the events to balance the ratio?
3. As the dataset is created from the closed youtube captions. Are you going to open-source it?

**Ethical Concerns:**

["NO or VERY MINOR ethics concerns only"]

**Final Justification:**

I thank the authors for their clarification and interesting work. I still have concerns about using the same model for generation and evaluation as this affects the quality of the dataset and lack of human annotation weakens the claim of output quality and correctness. Therefore, I will keep my score unchanged.

**Limitations:**

Yes

**Paper Formatting Concerns:**

Overall the paper is well written and easy to follow.

**Quality:**

3

**Strengths And Weaknesses:**

Strengths:
1. A novel dataset AudSem, creation of this dataset uses a multi-modal approach which makes the dataset more reliable.
2. The approach to semantic reasoning based architecture is novel.
3. Evaluates on most of the state-of-the art audio language models.

Weaknesses:
1. Same LLM used for caption generation and validation. In my view, this is not a good practice to validate captions, LLM will mark it's own generation as correct most of the time.

---

> ### Author Rebuttal · Authors · 2025-07-31
>
> We would like to thank the reviewer for their thorough feedback on our paper. We appreciate the recognition of our work's strengths, including the novelty of the dataset, the approach to semantic reasoning and the comprehensive evaluation on most of the state-of-the-art audio language models.
>
> > Same LLM used for caption generation and validation. In my view, this is not a good practice to validate captions, LLM will mark it's own generation as correct most of the time.
>
> We acknowledge this limitation. Our decision was based on two practical considerations:
> 1. This was the best open-source model available that could fit on one H100 GPU at the time based on benchmarks.
> 2. Resource optimization - using the same model allowed us to efficiently redo the judging and recaptioning repeatedly in one go without deploying additional GPU resources, which was crucial given the massive scale of our dataset (~7M).
>
> > Could you elaborate more on how captions of larger audios are generated?
>
> We deliberately limited our dataset to audios of maximum 30 seconds duration. This constraint was chosen to align with the capabilities of the Qwen2Audio model encoder used in our preliminary experiments. Qwen2Audio has a maximum input capacity of 3000 frames (at 10ms/frame = 30 seconds). Our model AudSemThinker is capable of converting 1 second of audio into 25 tokens and has a context window of 32,768 tokens, meaning 32768/25 = 1310.72 seconds. Theoretically, it can label up 21.5 minutes of audio, depending on the accompanying text prompt length.
>
> > I am curious how is the performance on music dataset so good as the distribution of music in the dataset characteristics chart shows the dataset contains more of sound events. Would also be interested to know more about the dataset distribution for sound events. Did you filter out the events to balance the ratio?
>
> In our filtered dataset (the final version used for training AudSemThinker), music captions actually constituted the majority compared to human sounds and other categories. It's worth noting that in the Audioset ontology, human sounds are subdivided into 9 subclasses (with speech being just one of them), while music has 5 subclasses. Music captions being the majority of the data, together with a specific Music captioning model for data generation (LP-MusicCaps), likely contributed to the strong performance on music-related tasks. We did not balance the ratio of the events.
>
> > As the dataset is created from the closed youtube captions. Are you going to open-source it?
>
> Yes, we have already open-sourced the dataset on the HuggingFace Datasets platform for easy access and download. Due to NeurIPS anonymity requirements, we cannot provide the direct link in this response, but it is publicly available.
>
> We thank the reviewer again for their valuable feedback.

---

> > ### Comment · Reviewer_nCca · 2025-08-08
> >
> > I appreciate the authors’ clarifications and responses. My assessment remains the same, and I will retain my current scores.

---

### Official Review · Reviewer_Xk32 · 2025-07-03

**Clarity:** 4
**Significance:** 4
**Originality:** 2
**Rating:** 4
**Confidence:** 5

**Summary:**

This paper proposes AUDSEMTHINKER, an audio-language model that enhances audio understanding and reasoning through structured semantic reasoning inspired by human cognitive perception of sounds.
 The authors also introduce AUDSEM, a new large-scale dataset curated from YouTube closed captions, designed to be more diverse and to minimize overlap with existing datasets.
The AUDSEMTHINKER model is trained using both supervised fine-tuning and reinforcement learning, taking advantage of semantic descriptors (such as “who, what, how, and where/when”).
Experiments demonstrate that their approach achieves good performance on multiple audio-language benchmarks, showing improved interpretability and reasoning over the semantics of sound.

**Questions:**

How did the authors extract the audio segments from YouTube videos? Since video data is typically sparsely sampled, how did the pipeline ensure that the extracted audio clips are aligned with the most relevant or “key” audio events described in the captions? Could the authors clarify the process by which they locate and select these important audio segments?

**Ethical Concerns:**

["NO or VERY MINOR ethics concerns only"]

**Final Justification:**

Thanks to the authors for the response. Regarding my question on the extraction of relevant sound information from YouTube videos, I have read Appendix C.1 of the appendix and find the authors’ approach to be reasonable and well thought out, where the process ensures precise alignment between captions and audio segments. Based on this clarification, I will raise my score. I also appreciate the detailed explanations provided in the Appendix C.1, and I encourage the authors to open source the dataset, as it would be a valuable resource for the research community.

**Limitations:**

See above. In my opinion, the evaluation (on MMAR) and performance improvement (compared with Omni-R1) are limited.

**Quality:**

2

**Strengths And Weaknesses:**

**Strengths**

1. The paper introduces a new audio-language model, based on structured semantic reasoning inspired by human auditory cognition (“who/what/how/when/where”). It also presents a new, diverse dataset (AUDSEM) created via a rigorous multi-modal, multi-stage pipeline, with careful filtering to reduce data overlap and leakage. In my opinion, the data is of great value.
2. The work is well-documented, highly reproducible, and open-source commitments are made.

**Weaknesses**

1. The paper does not report results on the MMAR[1] benchmark, which limits the completeness of its evaluation on the core idea (thinking process and audio reasoning).
2. There is no comparison with the recent Omni-R1[2] model, which achieves strong performance on MMAU (>70) using only answer reward with GRPO. This makes it unclear how the proposed approach stands relative to the state-of-the-art models.
3. There appears to be an inconsistency between Table 3 (SFT + AUDSEMTHINKER-QA (ours)) and Table 4 (SFT + Full + semantic + LoRA) results for MMAU Test-Mini; the numbers do not match up as expected and should be clarified.

[1] MMAR: A Challenging Benchmark for Deep Reasoning in Speech, Audio, Music, and Their Mix
[2] Omni-R1: Do You Really Need Audio to Fine-Tune Your Audio LLM?

---

> ### Author Rebuttal · Authors · 2025-07-31
>
> We would like to thank the reviewer for their thorough feedback on our paper.
>
> We are pleased that the reviewer recognized several strengths in our work, including:
> The novelty of our audio-language model with structured semantic reasoning inspired by human auditory cognition. The value of our AudSem dataset, which was created through a rigorous multi-modal, multi-stage pipeline. The documentation quality, reproducibility, and open-source commitment of our work.
>
> > The paper does not report results on the MMAR[1] benchmark, which limits the completeness of its evaluation on the core idea (thinking process and audio reasoning).
> There is no comparison with the recent Omni-R1[2] model, which achieves strong performance on MMAU (>70) using only answer reward with GRPO. This makes it unclear how the proposed approach stands relative to the state-of-the-art models.
>
> Both MMAR and Omni-R1 are valuable contributions to the community. However, the release of MMAR on May 19, 2025 occurred after the NeurIPS full paper deadline on May 15, 2025, and the release of Omni-R1 on May 14, 2025, occurred after the NeurIPS abstract submission deadline on May 11 2025. Therefore, these works could not possibly be included in our paper. We commit to incorporating this benchmark and model in future work.
>
> > There appears to be an inconsistency between Table 3 (SFT + AUDSEMTHINKER-QA (ours)) and Table 4 (SFT + Full + semantic + LoRA) results for MMAU Test-Mini; the numbers do not match up as expected and should be clarified.
>
> We appreciate the reviewer pointing out the apparent inconsistency between Tables 3 and 5. To clarify: in the Table 5 ablation study, each model was trained for exactly one epoch with the same amount of either semantic or simple data, taking approximately 12 hours on 1 GPU. This was done to ensure fair comparison across ablations. In contrast, our final AudSemThinker-QA model reported in Table 3 was trained for 3 epochs (totaling 36 hours on 1 GPU). This additional 24 hours of training resulted in the performance difference noted by the reviewer.
>
> > How did the authors extract the audio segments from YouTube videos? Since video data is typically sparsely sampled, how did the pipeline ensure that the extracted audio clips are aligned with the most relevant or “key” audio events described in the captions? Could the authors clarify the process by which they locate and select these important audio segments?
>
> Our YouTube closed caption dataset included precise "start" and "end" timestamps for each caption segment, measured as time intervals from the beginning of each video. We leveraged the open-source tool "yt-dlp" with its "–download-sections" parameter to efficiently extract only the specific audio segments corresponding to these timestamp boundaries. This targeted extraction approach was systematically applied across our entire dataset comprising 6,965,224 video clips, which ensured precise temporal alignment between captions and their corresponding audio content. For comprehensive technical details of this extraction pipeline, including preprocessing steps and quality verification, we refer the reviewer to Section C.1 in the Appendix.
>
> We thank the reviewer again for their valuable feedback.

---

> > ### Comment · Reviewer_Xk32 · 2025-08-06
> > **Re to rebuttal**
> >
> > Thanks to the authors for the response. Regarding my question on the extraction of relevant sound information from YouTube videos, I have read Appendix C.1 of the appendix and find the authors’ approach to be reasonable and well thought out, where the process ensures precise alignment between captions and audio segments. Based on this clarification, I will raise my score. I also appreciate the detailed explanations provided in the Appendix C.1, and I encourage the authors to open source the dataset, as it would be a valuable resource for the research community.

---

> > ### Comment · Reviewer_Xk32 · 2025-08-07
> > **More question for extended discussion period.**
> >
> > I have again reviewed the Omni-R1 work and noticed that they achieve better results than your method while using less data, simpler filtering engineering, and a simpler algorithm. Could you explain where this performance gap might stem from? I would appreciate your response on this issue.

---

> > > ### Author Response · Authors · 2025-08-07
> > > **Reponse to Reviewer Xk32**
> > >
> > > We thank the reviewer for their question regarding Omni-R1 and our work and we appreciate the opportunity to clarify several important distinctions that may contribute to the observed performance differences.
> > >
> > > > I have again reviewed the Omni-R1 work and noticed that they achieve better results than your method while using less data, simpler filtering engineering, and a simpler algorithm. Could you explain where this performance gap might stem from? I would appreciate your response on this issue.
> > >
> > > A consideration is the versioning of the MMAU benchmark. Our results were evaluated on the original MMAU benchmark, while some of Omni-R1's results were reported on the newer version released on May 25th, 2025, after the NeurIPS deadline. When comparing the performance of Omni-R1 to our performance in the old version of the benchmark, Omni-R1 outperforms our model, except for that our best supervised model achieves 76.65% on music data, outperforming Omni-R1's 74.3% on the same subset. This advantage stems from our dataset's amount of music examples.
> > >
> > > Our approach differs methodologically as well. While Omni-R1 directly generates answers, our model incorporates a semantic descriptor stage and reasoning process, which analyzes sounds in detail by identifying objects, actions, and contextual elements before answering. This additional complexity serves our research goals but does create implementation challenges, including the need for a dual-weighted reward function balancing both formatting and accuracy during training. Omni-R1 on the other hand only focused on accuracy, resulting in no reasoning step at all.
> > >
> > > An important point is regards dataset composition. We identified potential data contamination concerns with Omni-R1's training data. Approximately 31.79% of the MMAU benchmark originates from AudioSet (see the MMAU paper Appendix E), which has substantial overlap (36.81%) with VGGSound, the dataset used to train Omni-R1 (AVQA being based on VGGSound). Our dataset was deliberately constructed to have zero overlap with VGGSound, potentially trading some performance for scientific correctness and cleaner evaluation.
> > >
> > > Since our dataset was deliberately formed with the goal in mind to minimize overlap the most with respect to existing YouTube datasets, it required more extensive data filtering and cleaning steps to create a high-quality, purely audio-focused dataset. This results in having a data acquisition process that also involves more steps.
> > >
> > > Another consideration is that AVQA, being an audio-visual dataset, used in Omni-R1's training, may introduce visual elements that provide an unfair advantage when evaluating on audio-only tasks. Our approach consciously avoided this potential bias by specifically only letting the LLM generate auditory descriptions.
> > >
> > > We believe these factors collectively explain the apparent simplicity-performance trade-off observed between the two approaches. Our method prioritizes comprehensive sound understanding with explicit reasoning steps, while Omni-R1 takes a more direct path to answer prediction.

---

### Official Review · Reviewer_vgMe · 2025-07-07

**Clarity:** 3
**Significance:** 2
**Originality:** 2
**Rating:** 3
**Confidence:** 4

**Summary:**

The authors address the problem of improving the reasoning capability of audio language models on understanding semantics of sounds focusing mostly on acoustic and musical sounds (i.e, focus is not speech recognition). They propose a new dataset called AudSem and use this to further train the Qwen2.5-Omni 7B model using Parameter Efiicient Fine tuning (PEFT) with Low Rank Adapters (LoRA). Two training paradigms are explored - Supervised Fine-tuning (SFT) and Reinforcement Learning with Group Relative Policy Optimization (GRPO) to see which paradigm can extract the most out of their dataset. Performance of these models is assessed on publicly available benchmarks - MMAU and AudioBench.

Overall the most important contribution from this work is the AudSem Dataset and the necessary evaluation to show its utility. This dataset is mined from Youtube. this dataset is then filtered for sound and music descriptors with square brackets in closed captions and corresponding video and audio snippets are extracted. These extracted snippets are then labelled for sound semantics using expert audio and video classification models. A large Multimodal LLM (Qwen2.5 72B Instruct) is used to generate a caption by consuming the semantic information from both video and audio. Finally this dataset is further filtered using CLAP model to ensure relevance of generated semantic descriptions with the original labels.

**Questions:**

Raised in the Strengths and Weakness section.

**Ethical Concerns:**

["NO or VERY MINOR ethics concerns only"]

**Limitations:**

Societal Impact has not been discussed. At a cursory level the limitations of all underlying expert models used in data curation and training would hold as limitations of this model. Further it would be good to bring out known limitations of the dataset and discuss whether quality or quantity is helping in improving the reasoning capability of the trained models.

**Quality:**

2

**Strengths And Weaknesses:**

Strengths:
1. A sound pipeline to process and construct the dataset with minimal manual intervention using both video and audio modalities.
2. Paper is well written, easy to understand and widely accessible.
3. The curated dataset has minimal overlap with existing popular audio language datasets.
4. Interesting to see the results of SFT vs GRPO and to know that there is no clear winner.

Weaknesses:
1. The dataset has mostly been labelled using expert models - these expert models such as BEATs, LP-MusicCaps are far from being perfect. It is also not clear which specific version of BEATs the authors are using and it is possible that this model is overfitted on to AudioSet. I am also guessing that the authors used this model to generate the dataset characteristics shown in Figure 3. It is not clear what further analysis or cleaning the authors did to verify the quality of the labels from BEATs. A similar limitation holds for CLAP models and using this to filter the dataset is also problematic as there will be domain mismatch in the language of closed captions and that from Qwen2.5 72B model. It is not clear if CLAP's text encoder is capable of such generalization.
2. It is not clear if AudSem is a high quality dataset or simply a large enough dataset which has sufficient quality to help existing models do better on MMAU and AudioBench datasets. It this is the case, the authors would need to show on a few other existing models ( at least one more other than Qwen2.5 models) that similar gains can be observed on these benchmarks.

---

> ### Author Rebuttal · Authors · 2025-07-31
>
> We would like to thank the reviewer for their thorough feedback on our paper.
>
> We are pleased that the reviewer acknowledges the strengths of our paper, particularly:
> The sound pipeline to process and construct the dataset with minimal manual intervention. The clarity and accessibility of the paper. The minimal overlap of our dataset with existing audio language datasets, and the interesting comparison between SFT and GRPO approaches.
>
> > The dataset has mostly been labelled using expert models - these expert models such as BEATs, LP-MusicCaps are far from being perfect. It is also not clear which specific version of BEATs the authors are using and it is possible that this model is overfitted on to AudioSet. I am also guessing that the authors used this model to generate the dataset characteristics shown in Figure 3. It is not clear what further analysis or cleaning the authors did to verify the quality of the labels from BEATs. A similar limitation holds for CLAP models and using this to filter the dataset is also problematic as there will be domain mismatch in the language of closed captions and that from Qwen2.5 72B model. It is not clear if CLAP's text encoder is capable of such generalization.
>
> We acknowledge the reviewer's concern about the use of expert models for dataset labeling. We deliberately chose a variety of expert models, including BEATs and LP-MusicCaps, because of their expertise in specific task niches. For BEATs, this is audio tagging. For LP-MusicCaps, this is music captioning. The tags generated by the BEATs expert model were indeed used to create the visualization of top-level taxonomic parents of Audioset (Figure 3; left). All models were among the state-of-the-art in their respective tasks at the time of writing this paper.
>
> Importantly, we used a total of 9 different classification and generative language models across 3 modalities (Image, Audio, and Video) to provide the LLM with diverse perspectives for generating captions. By leveraging multiple models simultaneously, we achieve a statistical averaging effect that mitigates individual biases.
>
> In our research, we found that the CLAP encoder demonstrated effective generalization capabilities despite the domain difference, primarily due to the concise nature of the original YouTube subtitle fragments; when a closed caption contained phrases like "music playing," only Qwen2.5-generated captions containing similar semantic content were retained. CLAP also enhanced quality control by eliminating outliers through embedding distance calculations from the centroid of all combined embeddings.
>
> > It is not clear if AudSem is a high quality dataset or simply a large enough dataset which has sufficient quality to help existing models do better on MMAU and AudioBench datasets. It this is the case, the authors would need to show on a few other existing models ( at least one more other than Qwen2.5 models) that similar gains can be observed on these benchmarks.
>
> We would like to highlight that our Qwen2-Audio finetuned model achieved state-of-the-art performance among other Qwen2-Audio finetuned models. We included this in our ablation study (Table 5 on page 9), which demonstrates that similar gains to Qwen2.5-Omni can be observed when finetuning from Qwen2-Audio. This suggests that our approach generalizes beyond a single model architecture.
>
> > Societal Impact has not been discussed. At a cursory level the limitations of all underlying expert models used in data curation and training would hold as limitations of this model. Further it would be good to bring out known limitations of the dataset and discuss whether quality or quantity is helping in improving the reasoning capability of the trained models.
>
> We acknowledge that we could have discussed societal impact and limitations more thoroughly. In Section B.2. in the appendix we discuss potential positive societal benefits including enhanced audio transcription for individuals who are deaf or hard of hearing, sophisticated monitoring systems for environmental sounds like avian populations, and automated closed-caption generation for multimedia.
> We saw quantity as the primary driver for improving capability of trained models, particularly for GPRO training which required at least 50k samples before stabilizing its reward function. In future work, we plan to conduct more extensive evaluations across additional models and provide a more detailed analysis of whether quality or quantity is the primary driver.
>
> Thank you again for your valuable feedback.

---

> > ### Comment · Reviewer_vgMe · 2025-08-05
> > **Thank you for the point-to-point response**
> >
> > I thank the authors again for their interesting work and for taking the time to respond to my comments. I think my main concern about this work is related to the quality of the resulting dataset. One way to address this concern is through human evaluation which is currently not done as far as I can tell. While the use of this data results in SoTA performance it is not clear if this is due to dataset quality or some other unintended effect. I would like to stick to my original rating.

---

> > > ### Author Response · Authors · 2025-08-05
> > > **Respond to the Reviewer**
> > >
> > > Thank you for your reponse. We appreciate your concerns regarding dataset quality and would like to clarify a few points:
> > >
> > > Our approach deliberately minimizes human intervention to ensure scalability. As explained in Section 3 and Appendix C, we implemented a rigorous filtering pipeline that reduced our initial 6,965,224 samples to just 213,908 (3.07% of the original dataset), retaining only the highest quality captions.
> > >
> > > While comprehensive human evaluation wasn't feasible, we did conduct manual review of approximately 100 examples to verify dataset quality, finding effective captioning and answering across all cases. Much more extensive human evaluation would have been prohibitively time-consuming given our resources.
> > >
> > > Regarding performance improvements, we evaluated on multiple benchmarks using training methods comparable to other published results to keep unintended other effects to a minimum. Our ablation studies with Qwen2-Audio demonstrate that the improvements generalize beyond a single model. While we acknowledge that LoRA training preserves some previous knowledge, the consistent performance gains across different models suggest our dataset is contributing valuable new information.
> > >
> > > We believe our pipeline's ability to achieve state-of-the-art results with minimal human intervention is precisely the contribution's strength, and that it offers an efficient approach to creating high-quality training data at scale.

---

### Note · Authors · 2025-08-13

We would like to sincerely thank all reviewers for their invaluable and constructive feedback throughout this process. In response to the discussions, we have undertaken several key actions and clarifications to strengthen the paper and address all remaining concerns:

- To fully respect creator rights and resolve data control issues, we have fundamentally revised our data distribution strategy. Our AudSem dataset will now be released as YouTube IDs with start/end timestamps, not raw audio files. This aligns our work with established best practices and ensures content can only be accessed if it is still available on the original platform. The dataset, model, and code are all explicitly licensed under CC-BY-NC-SA-4.0, which prohibits commercial use.
- To address concerns about dataset quality, we conducted a formal manual evaluation of 100 samples with three annotators, which achieved moderate-to-substantial inter-annotator agreement (e.g., Krippendorff's alpha = 0.538 for correctness). In response to an analysis by Reviewer 1TFH with a text-only model, we acknowledged the presence of language priors in our multiple-choice questions. We contextualized this as a known phenomenon in the field and clarified that our dataset's diverse composition also includes tasks such as open-ended Q&A, captioning, and creative writing that necessitate genuine audio understanding, which mitigates reliance on language-only strategies.
- We have expanded our evaluation to include the OmniBench benchmark, where our model shows competitive performance against other models of a similar size. Furthermore, we updated our comparisons against the latest SOTA models, including Gemini 2.5 Pro, which confirms our model's strong performance, especially in the music category. We also clarified that other key benchmarks (e.g., MMAR) were released after the NeurIPS submission deadline and have committed to including them in future work.

We are confident that these revisions and clarifications have resolved the reviewers' concerns and have significantly improved the manuscript. We believe our work represents a valuable contribution to the community.

---

### Decision · Program_Chairs · 2025-09-17

**Decision:**

Accept (poster)

**Comment:**

This paper introduces AudSemThinker, an audio-language model focused on structured reasoning, and a new dataset, AudSem, created via a multi-modal pipeline from YouTube data. The work aims to improve semantic understanding in audio models and provides a new resource designed to have minimal overlap with existing benchmarks.

The reviewers' assessments and author discussions highlight several key points:

Contributions: The new AudSem dataset and its carefully designed, scalable creation pipeline were recognized as valuable contributions to the community (vgMe, Xk32, nCca). The model's structured reasoning approach was also seen as novel (Xk32, nCca).

Rebuttal summary: The authors were responsive and addressed most of the major concerns raised.

* Ethics: Significant ethical issues regarding data copyright and consent (ER H1d2, ER kDEN) were resolved by changing the data distribution plan to release YouTube IDs with timestamps and adopting a non-commercial license.

* Baselines: Concerns about missing comparisons to recent state-of-the-art models and benchmarks (Xk32, 1TFH) were convincingly addressed by noting they were released after the NeurIPS deadline. The authors further strengthened their evaluation during the rebuttal by adding new experiments on OmniBench and against Gemini 2.5 Pro, demonstrating competitive performance.

* Dataset Quality: In response to concerns about the automated pipeline (vgMe, nCca, 1TFH), the authors provided results from a manual annotation study, including inter-annotator agreement statistics, which helped alleviate fears about data quality (1TFH).

* Language Bias: A potential language bias in the dataset was acknowledged, but the authors contextualized it as a known phenomenon and highlighted that the majority of their dataset consists of open-ended tasks that require genuine audio understanding (1TFH).

This paper presents a solid contribution in the audio-language space. The authors' diligence during the rebuttal period addressed critical concerns regarding ethics, evaluation, and dataset validity. While some reservations about the automated data generation process remain (vgMe, nCca), the positive assessments from other reviewers (Xk32, 1TFH), the value of the new dataset, and the model's competitive performance justify its acceptance in AC opinion.